# ASC oligomer favors caspase-1[CARD] domain recruitment after intracellular potassium efflux

Fátima Martín-Sánchez[1], Vincent Compan[2,3], Alejandro Peñín-Franch[1], Ana Tapia-Abellán[1], Ana I. Gómez[1], María C. Baños-Gregori[1], Florian I. Schmidt[4], and Pablo Pelegrin[1,5]

Signaling through the inflammasome is important for the inflammatory response. Low concentrations of intracellular K[+] are associated with the specific oligomerization and activation of the NLRP3 inflammasome, a type of inflammasome involved in sterile inflammation. After NLRP3 oligomerization, ASC protein binds and forms oligomeric filaments that culminate in large protein complexes named ASC specks. ASC specks are also initiated from different inflammasome scaffolds, such as AIM2, NLRC4, or Pyrin. ASC oligomers recruit caspase-1 and then induce its activation through interactions between their respective caspase activation and recruitment domains (CARD). So far, ASC oligomerization and caspase-1 activation are K[+]-independent processes. Here, we found that when there is low intracellular K[+], ASC oligomers change their structure independently of NLRP3 and make the ASC[CARD] domain more accessible for the recruitment of the pro-caspase-1[CARD] domain. Therefore, conditions that decrease intracellular K[+] not only drive NLRP3 responses but also enhance the recruitment of the pro-caspase-1 CARD domain into the ASC specks.

## Introduction

The nucleotide-binding domain and leucine-rich repeat-containing receptor with a pyrin domain 3 (NLRP3) inflammasome is a regulator of inflammation and immunity. The NLRP3 inflammasome is a multiprotein complex whose oligomerization occurs in response to multiple pathogen and non-infectious triggers, including situations where intracellular ion homeostasis is disturbed (Hafner-Bratkovič and Pelegrín, 2018; Próchnicki et al., 2016). Changes in intracellular K[+] concentration have been linked to NLRP3 activation and the release of IL-1β (Perregaux and Gabel, 1994; Petrilli et al., 2007). Compounds, ion channels, and cell processes that decrease intracellular K[+] concentration, such as P2X7 receptor activation, TWIK2 ion channel opening, cell swelling, pore-forming toxins, or selective K[+]-ionophores, all lead to NLRP3 activation (Compan et al., 2012b; Martín-Sánchez et al., 2017; Di et al., 2018; Di Virgilio et al., 2017; Próchnicki et al., 2016; Muñoz-Planillo et al., 2013). Although the mechanism behind NLRP3 oligomerization in response to K[+]-efflux is not well understood, it has recently been suggested that lowering intracellular K[+] rearranges the Golgi and causes NLRP3 to bind to negatively charged lipids on the Golgi, thus promoting NLRP3's activation and later interaction with never in mitosis A-related kinase 7 (NEK7; Chen and Chen, 2018; He et al., 2016; Shi et al., 2016). Also, a decrease in intracellular K[+] leads to a conformational change in NLRP3, thus favoring its activation (Tapia-Abellán et al., 2021). NLRP3 oligomer binds the adaptor ASC (apoptosis-associated speck-like protein with a caspase recruitment domain) and promotes ASC oligomerization in filaments that form large protein complexes called ASC specks. ASC oligomerization from NLRP3 oligomers occurs via their pyrin domain (PYD) and the recruitment of new ASC subunits into the ASC filament continues via PYD–PYD interactions (Lu et al., 2014; Cai et al., 2014; Schmidt et al., 2016). In these filaments, the ASC caspase recruitment domain (CARD) is exposed to the external side and promotes filament binding and recruitment of pro-caspase-1 via CARD–CARD interactions that activate caspase-1 within inflammasome complexes (Boucher et al., 2018; Schmidt et al., 2016; Dick et al., 2016; Sborgi et al., 2015). Active caspase-1 is then able to process proinflammatory cytokines of the interleukin (IL)-1 family and also induces their release via the processing of gasdermin D and the formation of plasma membrane pores (Broz et al., 2020). Uncontrolled gasdermin D plasma membrane pores will lead to a specific type of cell death termed pyroptosis, which

[1]Molecular Inflammation Group, Biomedical Research Institute of Murcia (IMIB), Murcia, Spain; [2]IGF, Univ. Montpellier, CNRS, INSERM, Montpellier, France; [3]Laboratory of Excellence in Ion Channel Science and Therapeutics (Labex ICST), Villeneuve d'Ascq, France; [4]Institute of Innate Immunity, Medical Faculty, University of Bonn, Bonn, Germany; [5]Department of Biochemistry and Molecular Biology B and Immunology, Faculty of Medicine, University of Murcia, Murcia, Spain.

Correspondence to Pablo Pelegrín: pablo.pelegrin@imib.es

F. Martín-Sánchez's current affiliation is Manchester Collaborative Centre for Inflammation Research, The University of Manchester, Manchester, UK. A. Tapia-Abellán's current affiliation is Interfaculty Institute for Cell Biology, Department of Immunology, University of Tübingen, Tübingen, Germany.

amplifies the inflammatory response by releasing intracellular content, including inflammasome oligomers (Broz et al., 2020; Franklin et al., 2014; Young et al., 2006a). In addition to the NLRP3 inflammasome, ASC may also oligomerize and activate caspase-1 in response to other inflammasome scaffolds, such as the AIM2, Pyrin, or NLRC4 (Hornung et al., 2009; Masters et al., 2016; Proell et al., 2013). The oligomerization of these inflammasomes is independent of the concentration of intracellular $K^+$, and sensors are instead promoted by the presence of cytosolic nucleic acids, flagellin or type III secretion system proteins, or RhoA GTPase inhibition (Hornung et al., 2009; Masters et al., 2016; Gao et al., 2016; Zhao et al., 2011). To date, the oligomerization of ASC has been thought to be independent of intracellular $K^+$ concentration. Here, we describe how the ASC speck changes its structure in conditions of low intracellular $K^+$ to make ASC$^{CARD}$ domains more accessible and to enhance the recruitment of pro-caspase-1$^{CARD}$ domains. Our results show that the structure of ASC oligomerized by inflammasome sensors other than NLRP3 can be affected by a decrease in intracellular $K^+$.

## Results

### NLRP3, but not ASC, oligomerize in response to $K^+$ efflux
The co-expression of the inflammasome proteins NLRP3, ASC, and pro-caspase-1 in HEK293 cells results in the functional reconstitution of the NLRP3 inflammasome able to activate caspase-1 in response to a decrease in intracellular $K^+$ induced by hypotonic cell swelling (Fig. 1 A), which confirms that HEK293 cells are a suitable cellular model for studying NLRP3 inflammasome activation (Compan et al., 2012b; Chen and Chen, 2018; Bartok et al., 2016; Compan and López-Castejón, 2016). Both the $K^+$ ionophore nigericin and cell swelling were found to induce a decrease in the intracellular $K^+$ concentration in the HEK293 cell system (Fig. S1 A). NLRP3 and ASC co-expression in HEK293 results in the co-oligomerization of these proteins only after nigericin or hypotonic stimulation (Fig. 1 B). As expected, the oligomerization of NLRP3 was due to $K^+$ efflux since its oligomerization was blocked when nigericin was added in a high $K^+$ buffer (Fig. S1, B and C). Also, as previously described (Young et al., 2006a; Tapia-Abellán et al., 2019a), the pathological mutation of NLRP3 D303N associated with cryopyrin-associated periodic syndromes (CAPS) resulted in the spontaneous oligomerization of NLRP3 when expressed in HEK293 cells without any stimulation (Fig. 1 C). This oligomerization was not affected by $K^+$ efflux, as nigericin or hypotonic stimulation did not affect basal NLRP3 D303N oligomers (Fig. S1 D). When ASC is co-expressed with the D303N mutated NLRP3, ASC assembly led to the convergence of NLRP3 oligomers into single structures per cell (Fig. 1 C).

The expression of low amounts of ASC alone in HEK293 was unable to trigger oligomerization in response to nigericin or hypotonic solution (Fig. 1 D), and NLRP3-deficient macrophages expressing ASC do not release IL-1β after nigericin stimulation (Fig. S1 E), thus confirming that NLRP3 is the sensor protein responsible for the oligomerization of functional inflammasomes

in response to $K^+$ efflux (Muñoz-Planillo et al., 2013). However, for higher expression levels of ASC alone, spontaneous oligomerization of ASC was observed in HEK293 (Fig. 1 D). These oligomers were functional as a platform for inducing the self-processing of caspase-1 and IL-1β (Fig. 1 E) and they recruit wild-type NLRP3 into the ASC specks without any cell stimulation that would decrease intracellular $K^+$ (Fig. 1 F). Therefore, ASC specks formed by high expression levels of ASC present a functional structure. To study the structure of these ASC oligomers, we did high-resolution imaging and found that ASC oligomers were filamentous in nature when both untagged and Luc-tagged ASC were expressed (Fig. 1 G), similar to what has been reported in other studies (Hoss et al., 2016). When ASC was expressed together with a nanobody targeting ASC$^{CARD}$ domain and impairing ASC$^{CARD}$ homotypic interactions (Schmidt et al., 2016), ASC oligomerization was found to result in filaments (Fig. 1 H).

### BRET assay reveals a structural change in ASC specks in response to $K^+$ efflux
Spontaneous ASC speck formation induced by high expression was not affected by tagging ASC with either YFP or Luc at C-terminus (Fig. S2, A and B). Co-expression of ASC-YFP and ASC-Luc results in ASC specks containing both types of tagged ASC (Fig. 2 A). Therefore, we used the BRET technique to monitor the distance between the donor Luc and the receptor YFP within the speck in real time. The BRET signal is higher when these two epitopes are closer to each other and is lower when they are further apart (Compan et al., 2012a). We found a specific BRET signal between ASC-Luc and ASC-YFP inside the specks (Fig. 2, B and C). After nigericin or hypotonic stimulation, the ASC speck BRET signal decreased (Fig. 2 D), and this decrease was blocked by high extracellular $K^+$ concentration (Fig. 2 E and Fig. S2 C), suggesting that the ASC speck, independently of NLRP3, undergoes a structural change caused by $K^+$ efflux. Co-expression of wild-type NLRP3 or D303N mutant increased the ASC speck BRET signal (Fig. 2 F) but did not affect the general profile of the ASC speck structural changes induced by $K^+$ efflux (Fig. 2 G). However, the presence of NLRP3 resulted in a faster and more pronounced decrease in the BRET signal after nigericin or hypotonic stimulation (Fig. 2 G).

### $K^+$ efflux-induced structural changes in the ASC speck promote differential ASC labeling with antibodies
The structural changes in the ASC speck due to $K^+$ efflux were also shown by immunofluorescence experiments and ASC staining with different antibodies. Using anti-Luc antibodies, the staining of the ASC speck that was formed by ASC-YFP and ASC-Luc was more intense after nigericin or hypotonic treatment (Fig. 3, A and B), whereas ASC-YFP fluorescence remained constant whatever the treatment (Fig. S3 A). Change in ASC-Luc staining was transient after nigericin treatment and stable after hypotonic stimulation (Fig. 3, A and B). The increase in ASC staining observed after nigericin treatment was blocked by a high $K^+$ extracellular solution (Fig. 3 C). These data are consistent with the BRET recording kinetics described above (Fig. 2 D). Similarly, in HEK293 overexpressing untagged mouse or human

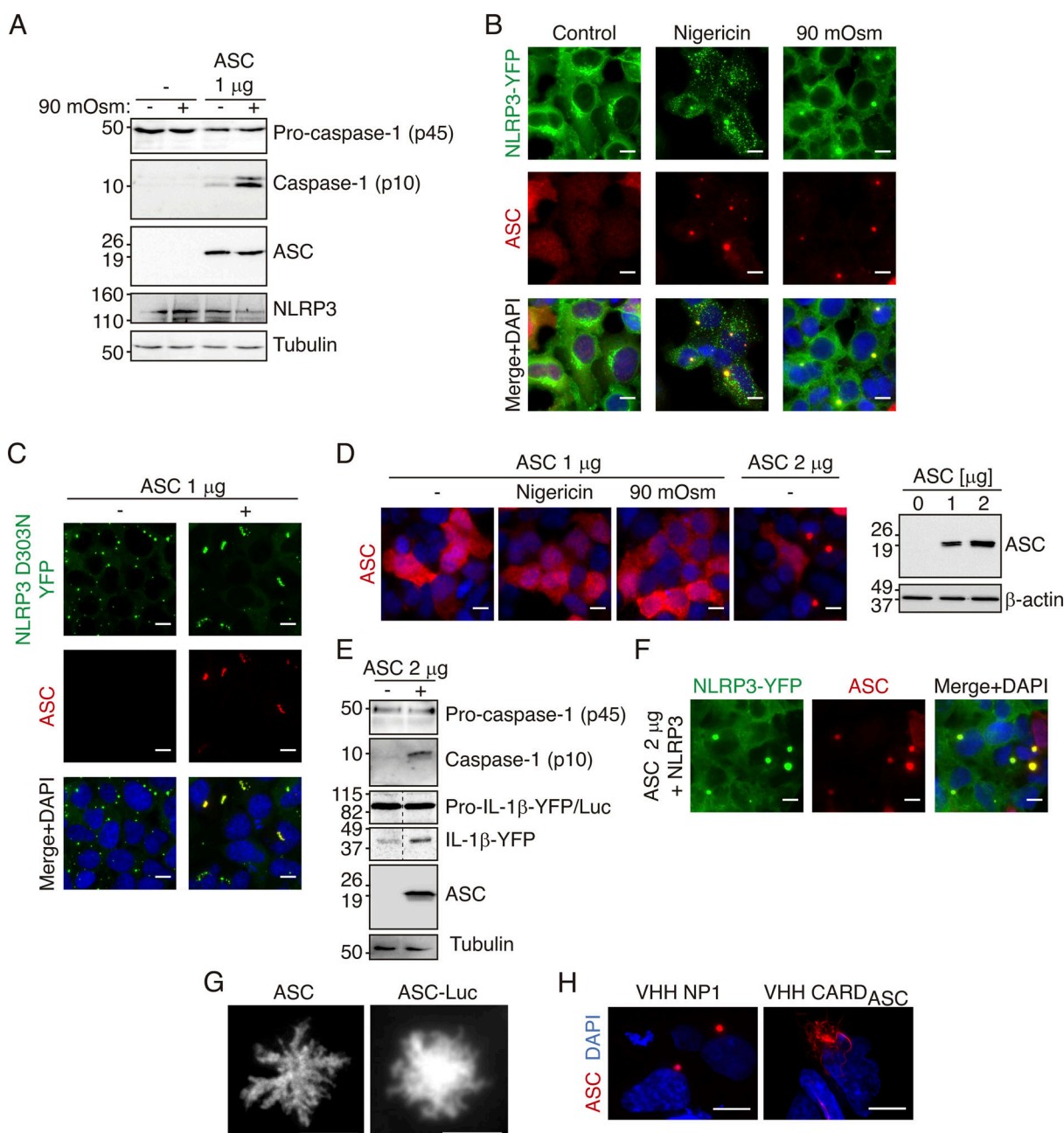

Figure 1. **K+ efflux induces NLRP3, but not ASC, oligomerization. (A)** Caspase-1, ASC, NLRP3, and tubulin immunoblot of HEK293 cells transfected with plasmids encoding for pro-caspase-1, NLRP3, and in the last two lanes with ASC (1 μg). After 16 h of transfection, cells were exposed to hypotonic buffer (90 mOsm) for 1 h as indicated before cell lysis. **(B)** Representative fluorescent photomicrographs of NLRP3-YFP HEK293 stable cell line transfected with a plasmid encoding for ASC (1 μg) and unstimulated (control) or stimulated with either nigericin (10 μM, 30 min) or hypotonic solution (90 mOsm, 1 h) as indicated. NLRP3-YFP was revealed by direct excitation of YFP (green), ASC was stained with polyclonal α-PYD$_{ASC}$ antibody (AF647, red), and nuclei were revealed with DAPI (blue); scale bar, 10 μm. **(C)** Representative fluorescent photomicrographs of NLRP3-YFP D303N HEK293 stable cell line transfected or not with a plasmid encoding for ASC (1 μg) as indicated. NLRP3-YFP D303N was revealed by direct excitation of YFP (green), ASC was stained with polyclonal α-PYD$_{ASC}$ antibody (AF647, red), and nuclei were revealed with DAPI (blue); scale bar, 10 μm. **(D)** Representative fluorescent photomicrographs of HEK293 cells transfected with a plasmid encoding for ASC (1 or 2 μg, as indicated). ASC was stained with polyclonal α-PYD$_{ASC}$ antibody (AF647, red) and nuclei were revealed with DAPI (blue); scale bar 10 μm (left panels). ASC and β-actin immunoblot of cell extracts (right). **(E)** Caspase-1, IL-1β, ASC, and tubulin immunoblot of HEK293 cells transfected with plasmids encoding for pro-caspase-1, pro-IL-1β, and ASC (2 μg). **(F)** Representative fluorescent photomicrographs of NLRP3-YFP HEK293 stable cell line transfected with a plasmid encoding for ASC (2 μg). NLRP3-YFP was revealed by direct excitation of YFP (green), ASC was stained with polyclonal α-PYD$_{ASC}$ antibody (AF647, red) and nuclei were revealed with DAPI (blue); scale bar 10 μm. **(G)** Deconvolved representative fluorescent photomicrographs of HEK293 cells transfected with a plasmid encoding for ASC (2 μg). ASC was stained with monoclonal α-CARD$_{ASC}$ antibody (AF647, red); scale bar, 10 μm. **(H)** Deconvolved representative fluorescent photomicrographs of HEK293 cells transfected with plasmids encoding for ASC-RFP (0.1 μg) and for a control nanobody (VHH NP1) or for a α-CARD$_{ASC}$ nanobody (VHH CARD$_{ASC}$). ASC-RFP was revealed by direct excitation of RFP (red), and nuclei were revealed with DAPI (blue); scale bar, 10 μm.

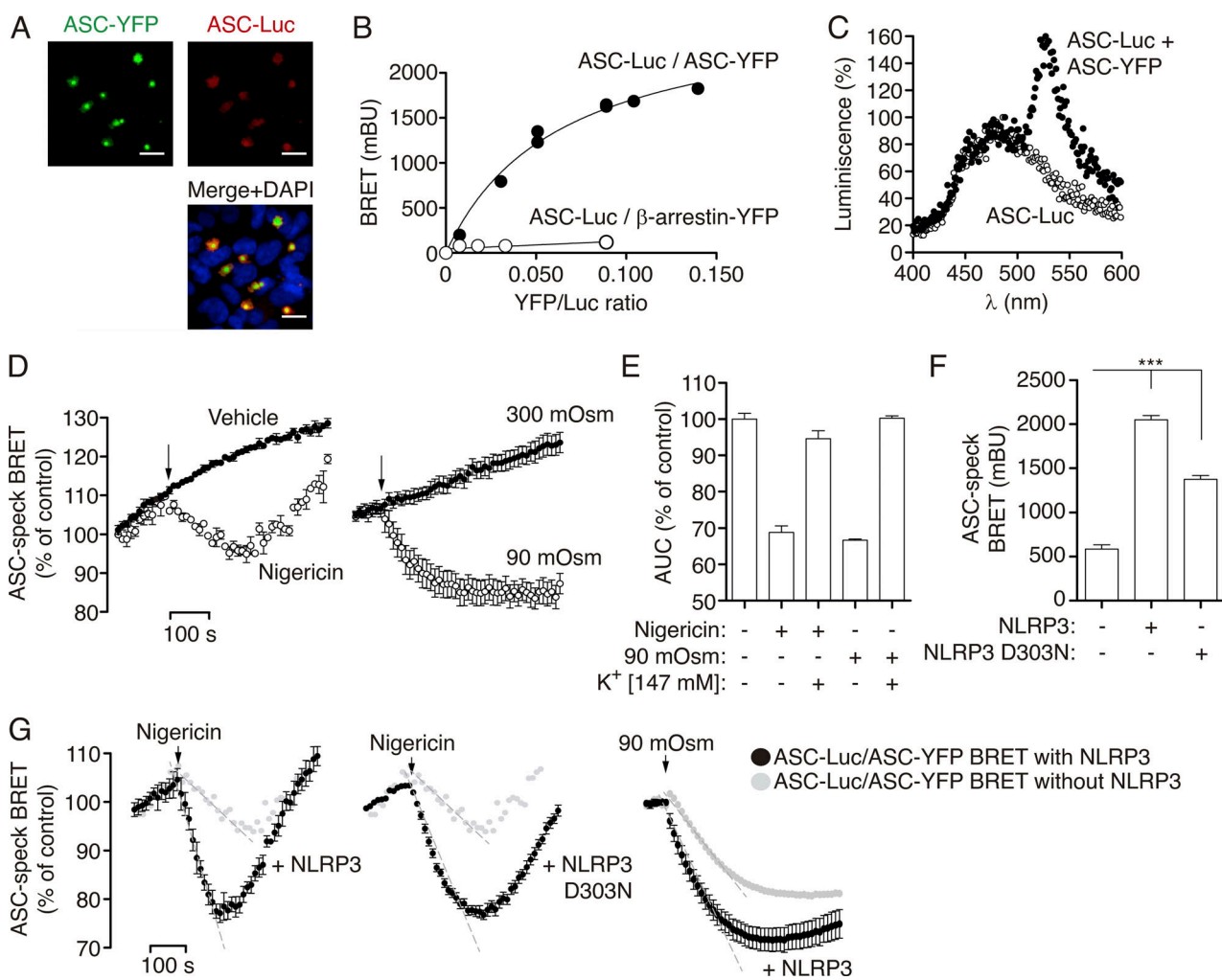

Figure 2. **BRET assays reveal a structural change in ASC specks in response to K⁺ efflux. (A)** Representative fluorescent photomicrographs of HEK293 cells transfected with plasmids encoding for ASC-YFP (0.2 µg) and ASC-Luc (0.1 µg). ASC-YFP was revealed by direct excitation of YFP (green), ASC-Luc was stained with α-Luc antibody (AF647, red), and nuclei were revealed with DAPI (blue); scale bar, 10 µm. **(B)** BRET saturation curves for HEK293 cells transfected with a constant concentration of ASC-Luc and increasing amounts of the BRET acceptor ASC-YFP (black circles) or β-arrestin-YFP (white circles). mBU, milliBRET units. **(C)** Emission spectra of HEK293 cells expressing ASC-Luc (white circles) or ASC-Luc with ASC-YFP (black circles) after adding coelenterazine h. Note the peak emission at 535 nm as the BRET among ASC-Luc and ASC-YFP inside the ASC speck. **(D)** Kinetic of net BRET signal in HEK293 cells transfected with the BRET donor ASC-Luc and the acceptor ASC-YFP in response to nigericin (left) or hypotonic solution (90 mOsm, right; n = 2–10 biological replicates). **(E)** Area under the curve (AUC) of the net BRET signal kinetic in HEK293 cells transfected with the BRET donor ASC-Luc and the acceptor ASC-YFP in response to nigericin or hypotonic solution (90 mOsm) recorded in a buffer containing 147 mM of KCl (n = 5–13 biological replicates). **(F)** BRET signal in HEK293 cells transfected with the BRET donor ASC-Luc and the acceptor ASC-YFP in the presence of NLRP3 wild-type or D303N as indicated (n = 9–19 biological replicates). **(G)** Kinetic of net BRET signal in HEK293 cells transfected with the BRET donor ASC-Luc and the acceptor ASC-YFP in the presence of NLRP3 wild-type or D303N as indicated, in response to nigericin (left two panels) or hypotonic solution (90 mOsm, right panel); light gray dots represent the BRET signal for ASC-Luc and ASC-YFP in the absence of NLRP3 expression (n = 3–10 biological replicates). For D-G, data are represented as mean and error bars represent ± SEM; Kruskal–Wallis test with Dunn's multiple-comparison post-test was used in F, ***P < 0.005.

ASC, using an anti-ASC antibody against the CARD domain, we observed changes in ASC staining after hypotonic or nigericin stimulation (Fig. 3, D–F). However, and differentially to mouse ASC, human ASC staining decreased after 20 min of hypotonic stimulation but was still significantly higher than in control conditions (Fig. 3, B–E). This could be due to differences between mouse and human ASC speck structure or due to the specificity of the antibody against the human-ASC^CARD domain, which was specifically recognized in oligomeric ASC but not in soluble ASC, either overexpressed in HEK293 cells or endogenously expressed in mouse macrophages (Fig. S3, B and C). These data

suggest that upon K⁺ efflux, the ASC^CARD domain is more accessible to antibody staining.

Without any stimulation, the antibody against the ASC^CARD was unable to recognize soluble or oligomeric mouse ASC tagged at Ct with YFP (Fig. 4 A and Fig. S3 D). However, after nigericin or hypotonic treatment, mouse ASC-YFP specks were stained (Fig. 4, A and B) and this staining was completely abolished by high extracellular concentration of K⁺ (Fig. 4 B). Of note, and contrary to the mouse ASC, the BioLegend antibody against the ASC^CARD can stain both soluble and oligomeric human ASC (Table S1). In contrast to the staining of the CARD domain, an

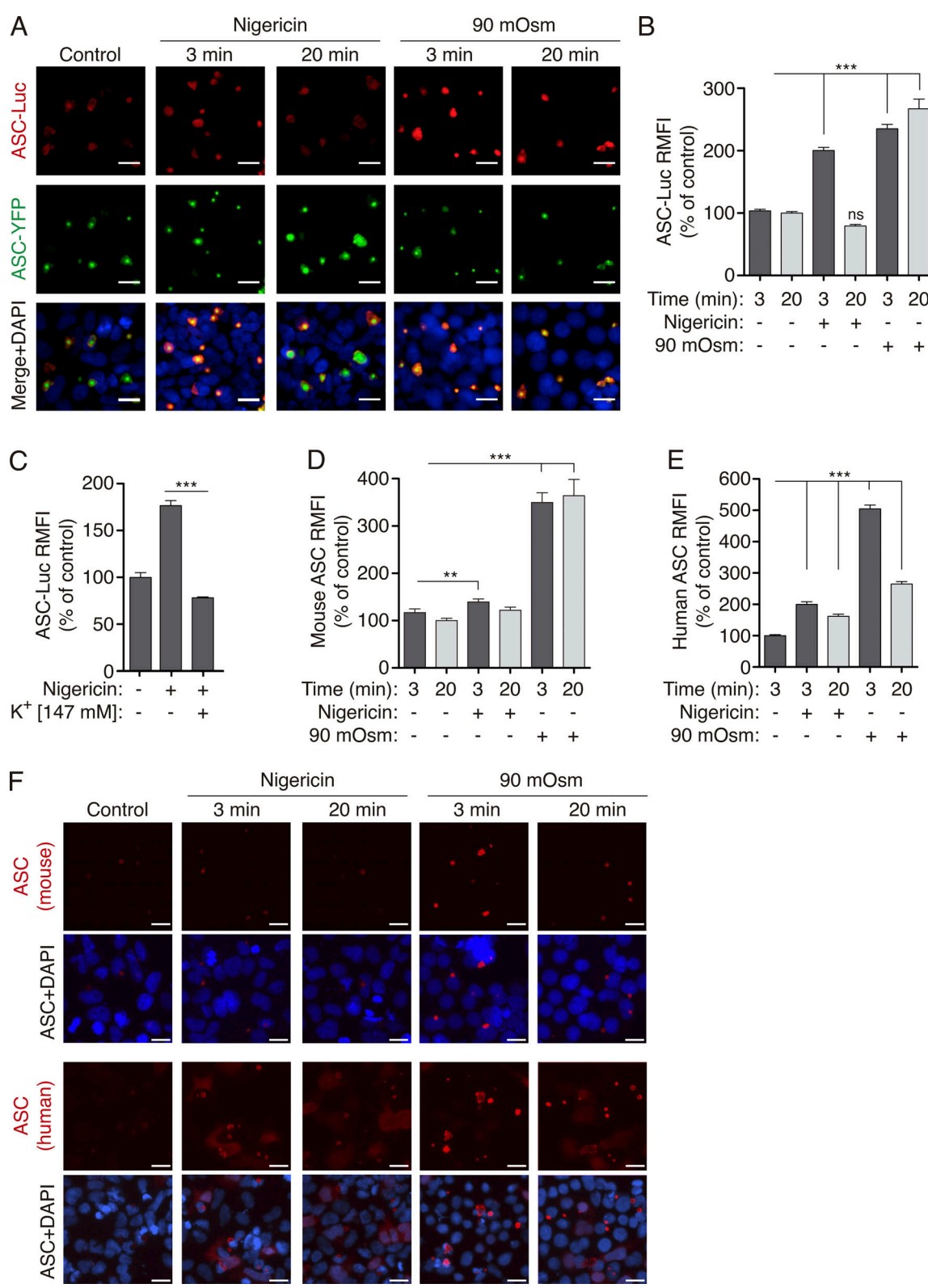

Figure 3. **ASC speck structural change in response to K⁺ efflux is revealed with immunofluorescence. (A)** Representative fluorescent photomicrographs of HEK293 cells transfected with plasmids encoding for ASC-YFP (0.2 µg) and ASC-Luc (0.1 µg), unstimulated (control), or stimulated with nigericin (10 µM) or hypotonic solution (90 mOsm) for the indicated time. ASC-YFP was detected by the YFP fluorescence (green), ASC-Luc was stained with α-Luc antibody (AF647, red), and nuclei were revealed with DAPI (blue); scale bar, 10 µm. **(B)** Relative mean fluorescence intensity (RMFI) of ASC-Luc oligomers from HEK293 transfected and stimulated as in A. Data from $n$ = 3 independent experiments and quantification of a total of 2,084 ASC specks. **(C)** RMFI of ASC-Luc oligomers from HEK293 transfected and stimulated as in A but when indicated in the presence of a buffer containing 147 mM KCl. Data from $n$ = 3 independent experiments and quantification of a total of 1,278 ASC-specks. **(D and E)** RMFI of ASC oligomers from HEK293 transfected with a plasmid encoding for untagged

mouse ASC (2 µg, D) or untagged human ASC (1 µg, E) and stimulated as in A. ASC was labeled with monoclonal α-CARD$_{ASC}$. Data from $n$ = 2–3 independent experiments and quantification of a total of 1,323 (D) and 1,331 (E) ASC specks. **(F)** Representative fluorescent photomicrographs of cells with ASC specks quantified in D and E. ASC was stained with monoclonal α-CARD$_{ASC}$ antibody (AF647, red) and nuclei was revealed with DAPI (blue); scale bar, 20 µm. For B–E data are represented as mean and error bars represent ± SEM; Kruskal–Wallis test with Dunn's multiple-comparison post-test was used in B–E, ***P < 0.001; **P < 0.01; ns, not significant (P > 0.05) difference.

antibody against the ASC$^{PYD}$ domain was able to detect both soluble and oligomeric ASC either untagged (Fig. 1 D) or tagged with YFP at Ct (Fig. S3 B), and this staining was not affected by changes in intracellular K$^+$ concentration (Fig. 4 B). These data suggest that K$^+$ efflux induces a change in ASC speck structure that makes the ASC$^{CARD}$ domain, but not the ASC$^{PYD}$ domain, more accessible to antibody staining.

To identify if the change observed in the structure of the ASC speck was due to changes in the PYD–PYD oligomerization

filament or the CARD–CARD interaction of different filaments, we used a nanobody that binds to the ASC$^{CARD}$ domain and impairs the ASC$^{CARD}$ homotypic interactions (Schmidt et al., 2016). In cells coexpressing ASC-YFP and ASC-Luc BRET sensors, this resulted in ASC filamentous structures containing both types of ASC (Fig. 4 C). BRET recording showed that ASC filaments presented a lower BRET signal when compared with the BRET signal from the ASC speck co-expressed with a control nanobody (Fig. 4 D), suggesting that oligomerization of ASC into

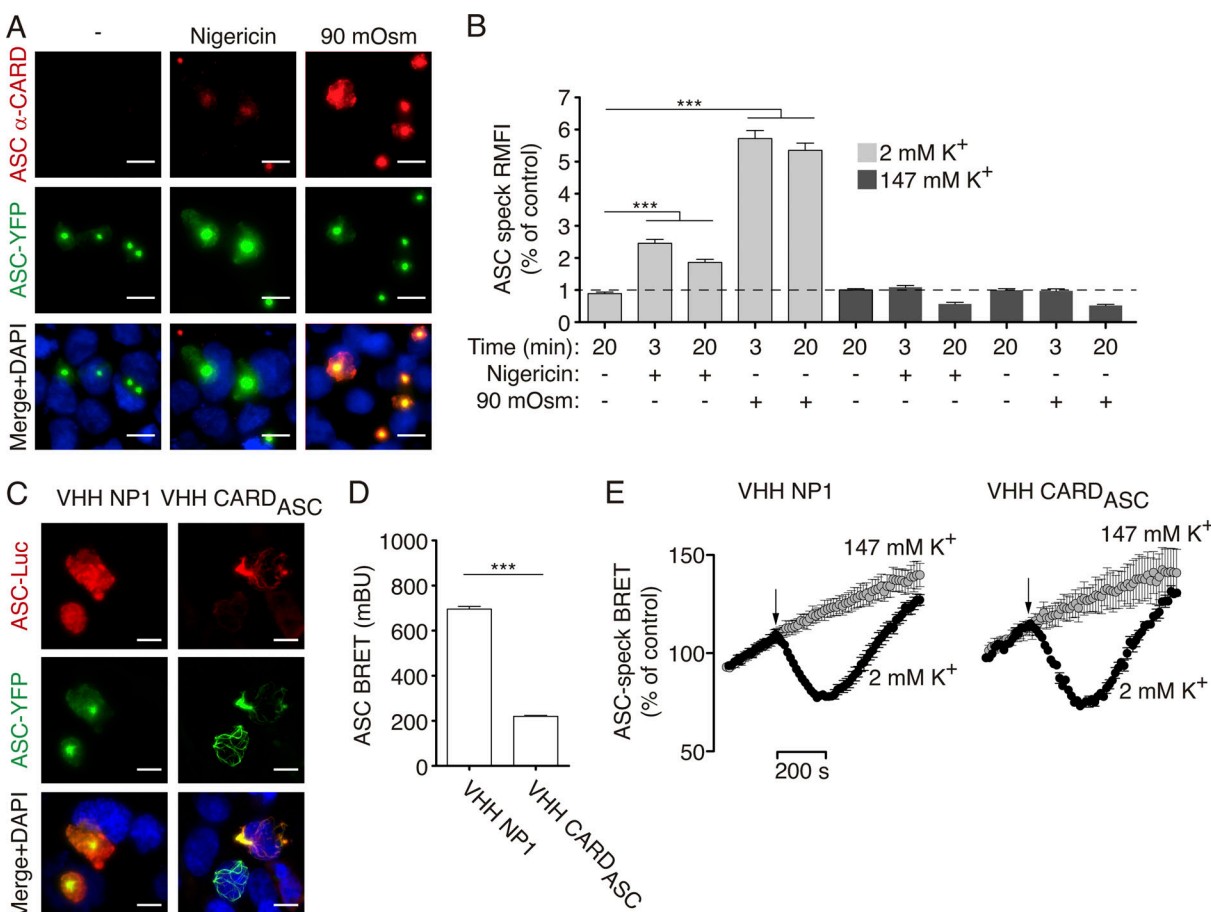

Figure 4. **ASC speck structural change is differentially labeled with antibodies. (A)** Representative fluorescent micrographs of HEK293 cells transfected with a plasmid encoding for ASC-YFP (0.2 µg), unstimulated (control), or stimulated with nigericin (10 µM) or hypotonic solution (90 mOsm) for 3 min. ASC was detected by the YFP fluorescence (green) and stained with monoclonal α-CARD$_{ASC}$ antibody (AF647, red), and nuclei were revealed with DAPI (blue); scale bar, 10 µm. **(B)** Relative mean fluorescence intensity (RMFI) of ASC-YFP oligomers stained with monoclonal α-CARD$_{ASC}$ antibody from HEK293 transfected and stimulated as in A, but in a buffer with 2 mM (light gray) or 147 mM (dark gray) KCl. Data from $n$ = 2 independent experiments and quantification of a total of 1,103 ASC-specks (2 mM KCl), 370 ASC-specks (nigericin with 147 mM KCl), or 465 ASC-specks (90 mOsm with 147 mM KCl). **(C)** Representative fluorescent photomicrographs of HEK293 cells transfected with plasmids encoding for ASC-YFP (0.2 µg), ASC-Luc (0.1 µg), and for either control nanobody (VHH NP1, 0.5 µg) or α-CARD$_{ASC}$ nanobody (VHH CARD$_{ASC}$, 0.5 µg). ASC-YFP was revealed by direct excitation of YFP (green), ASC-Luc was stained with α-Luc antibody (AF647, red) and nuclei were revealed with DAPI (blue); scale bar, 10 µm. **(D)** BRET signal in HEK293 cells transfected as in C. Data from $n$ = 3 independent experiments. **(E)** Kinetic of net BRET signal in HEK293 cells transfected as in C in response to nigericin in normal extracellular K$^+$ buffer (2 mM K$^+$) or high K$^+$ buffer (147 mM K$^+$). Data from $n$ = 4 independent experiments. For B, D, and E, data are represented as mean, and error bars represent ± SEM; Kruskal–Wallis test with Dunn's multiple-comparison post-test was used in B, ***P < 0.001; $t$ test was used in D, ***P < 0.001.

a speck led to a stronger BRET signal compared with that of the ASC filament. After nigericin treatment, we found a similar change in the BRET signal that was abrogated by using a high extracellular K$^+$ solution (Fig. 4 E), suggesting that the structural change in the ASC filament is similar to that of the full ASC speck. This suggests that the oligomerization of ASC by the ASC$^{PYD}$ domain undergoes structural quaternary changes after K$^+$ efflux that increase the accessibility of the ASC$^{CARD}$ domain.

### K$^+$-efflux-induced structural changes to the ASC specks lead to higher recruitment of pro-caspase-1$^{CARD}$

Our results lead us to hypothesize that the change in the ASC speck structure exposes the ASC$^{CARD}$ domain in low K$^+$ intracellular environments, which may result in better recruitment of pro-caspase-1. To test this hypothesis, we stained ASC specks with cytosolic extracts containing either soluble pro-caspase-1$^{CARD}$ domain tagged with EGFP (unpublished data, F.I. Schmidt lab), which would be recruited by ASC$^{CARD}$ interactions, or soluble ASC-RFP, which would be recruited mainly by ASC$^{PYD}$ interactions, but also by ASC$^{CARD}$ interactions. By detecting the GFP of the pro-caspase-1$^{CARD}$ domain and the RFP fluorescence of the ASC-RFP, we found that both had been recruited by already-formed ASC specks (Fig. 5 A). However, the ASC antibody against the ASC$^{CARD}$ domain failed to stain ASC-RFP recruited to ASC speck (Fig. 5 B) because this antibody does not recognize mouse ASC tagged on the Ct in the absence of CARD domain exposure (Fig. S3 A). When K$^+$ efflux was induced by hypotonic stimulation of HEK293 cells expressing untagged ASC specks, we found an increase in the recruitment of the pro-caspase-1$^{CARD}$ domain, but not in additional ASC-RFP subunits (Fig. 5, C and D). Staining of ASC with the anti-ASC$^{CARD}$ domain antibody showed a structural change in the ASC speck upon hypotonic stimulation that was further increased by the recruitment of ASC-RFP into the speck (Fig. 5 E). This suggests that the recruitment of new proteins via the ASC$^{CARD}$ domain is facilitated in situations of low intracellular K$^+$, but not the recruitment of new subunits of ASC, possibly via the ASC$^{PYD}$ domain.

Finally, to test if this mechanism could have a physiological role, we stimulated the Pyrin inflammasome with the toxin B of *Clostridium difficile* (TcdB), which resulted in the oligomerization of ASC by Pyrin assembly independently of LPS priming (Heilig and Broz, 2018). Caspase-1/11 deficient macrophages were used to prevent pyroptotic cell death upon Pyrin-induced ASC oligomerization, and by omitting LPS-priming, we ensured that NLRP3 inflammasome would not be activated in these experiments (Hornung and Latz, 2010). In this situation, TcdB, but not nigericin or hypotonic stimulation, was able to induce ASC speck formation (Fig. 6 A). Furthermore, nigericin and hypotonic stimulation were not able to alter the number of specking macrophages upon TcdB stimulation (Fig. 6 A). However, when nigericin or hypotonicity was applied after TcdB stimulation, the staining intensity in ASC specks determined by immunofluorescence using anti-ASC$^{CARD}$ antibody was significantly increased and blocked when a high extracellular K$^+$ solution was used (Fig. 6, B and C). Furthermore, cellular K$^+$ depletion induced by nigericin also increased the staining intensity of ASC

specks in caspase-1/11 deficient macrophages after the activation of the NLRC4 inflammasome while the NLRP3 inflammasome was blocked by MCC950 (Fig. 6 D). Despite the increased accessibility of the ASC$^{CARD}$ domain within the ASC speck when intracellular K$^+$ was decreased, we found that neither nigericin nor the use of a buffer with a high K$^+$ concentration was able to modulate IL-1β release from *Nlrp3$^{-/-}$* macrophages (Fig. S4). This could be because when IL-1β is detected in the supernatant there is already enough pyroptosis to saturate the system, and the modulation of the ASC speck by K$^+$ efflux and further increase of caspase-1 activation could not significantly change the final amount of released cytokine.

In conclusion, our study demonstrates that intracellular K$^+$ efflux modulates the ASC oligomer structure at the level of the ASC filament beyond the activation of the NLRP3 inflammasome, which exposes the ASC$^{CARD}$ domain and promotes the recruitment of pro-caspase-1$^{CARD}$ domain into the ASC speck.

## Discussion

We described in this study how the quaternary structure of the ASC oligomer changes after cellular treatment with the K$^+$ ionophore nigericin or during hypotonic-induced cell swelling. This structural change can already be observed in ASC filaments assembled by PYD–PYD interactions and results in a more exposed ASC$^{CARD}$ domain. This allows better recruitment of the pro-caspase-1$^{CARD}$ domain to the ASC speck.

The NLRP3 inflammasome sensor protein is the only inflammasome component described to be activated by a decrease in intracellular K$^+$ and therefore is the only inflammasome triggered by compounds and processes that induce a K$^+$ efflux from cells, such as specific K$^+$-ionophores, activation of ion channels, or pore-forming toxins (Hafner-Bratkovič and Pelegrín, 2018; Próchnicki et al., 2016; Muñoz-Planillo et al., 2013). Also, the activation of caspase-11 induces gasdermin D pyroptotic pores on the plasma membrane that induces efflux of K$^+$ and the downstream activation of the NLRP3 inflammasome (Kayagaki et al., 2011, 2015). We found that when intracellular K$^+$ decreases, the ASC speck changes its structure, which in turn promotes the recruitment of the pro-caspase-1$^{CARD}$ domain, suggesting that initial gasdermin D plasma membrane permeabilization induced by other K$^+$-independent inflammasomes affects the structure of the ASC specks, even if NLRP3 is not present or primed. Therefore, intracellular K$^+$ concentration regulates at different levels the inflammasome structure, including inflammasomes that are not directly triggered by a decrease in intracellular K$^+$, such as AIM2, NLRC4, or Pyrin (Hornung et al., 2009; Masters et al., 2016; Gao et al., 2016; Zhao et al., 2011). In particular, ASC specks triggered by the activation of the Pyrin inflammasome are also able to change their structure independently of NLRP3 when intracellular K$^+$ was decreased.

We previously described that the decrease in intracellular Cl$^-$ is important in inducing the oligomerization of ASC by NLRP3 activation (Green et al., 2018); however, under those conditions, the resulting ASC oligomers were unable to activate caspase-1, unless the cells also present an efflux of K$^+$ that allowed the interaction between NLRP3 and NEK7, which in turn activated

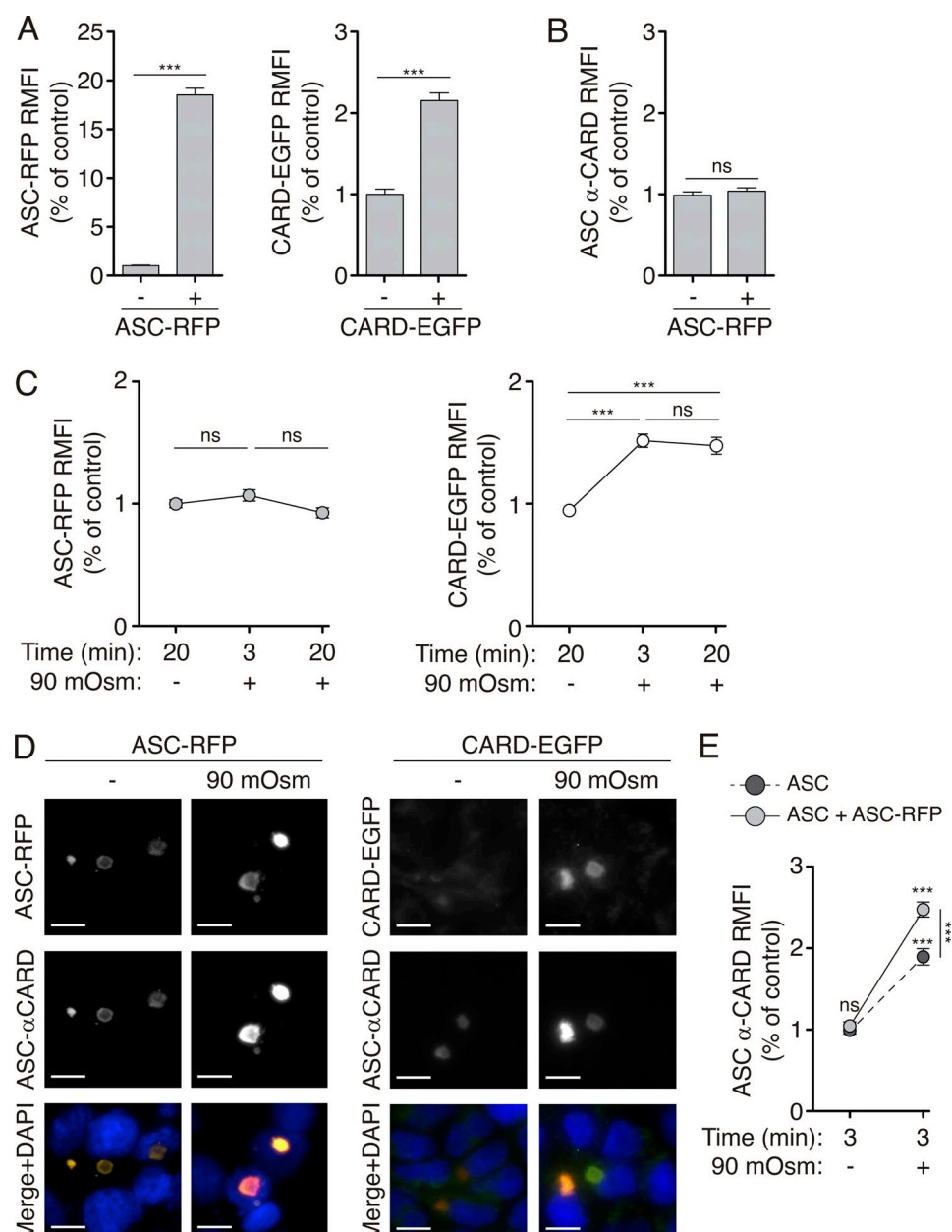

Figure 5. **ASC specks recruit more caspase-1$^{CARD}$ domain upon K$^+$ efflux. (A)** Relative mean fluorescence intensity (RMFI) of ASC-RFP (left panel) or CARD$_{Casp1}$-EGFP (right panel) oligomers from HEK293 transfected with untagged ASC (2 µg), then fixed and permeabilized and stained (+) or not (−) with soluble ASC-RFP (left panel) or CARD$_{Casp1}$-EGFP (right panel). Data from $n$ = 2–3 independent experiments and quantification of a total of 467 (left panel) and 258 (right panel) ASC specks. **(B)** RMFI of ASC specks stained with monoclonal α-CARD$_{ASC}$ antibody from HEK293 transfected as in A and treated (+) or not (−) with soluble ASC-RFP. Data from $n$ = 2–3 independent experiments and quantification of a total of 467 ASC specks. **(C)** RMFI of ASC specks present in cells transfected as in A and stained with soluble ASC-RFP (left panel) or CARD$_{Casp1}$-EGFP (right panel). Before fixation/staining cells were stimulated (+) or not (−) with hypotonic solution (90 mOsm) for the indicated times (control condition is 20 min with isotonic solution). Data from $n$ = 2–3 independent experiments and quantification of a total of 464 (left panel) and 586 (right panel) ASC specks. **(D)** Representative fluorescent photomicrographs of HEK293 cells transfected, treated, and stained as in C for 20 min. Nuclei were revealed with DAPI (blue); scale bar, 10 µm. **(E)** RMFI of the ASC speck intensity stained with α-CARD$_{ASC}$ antibody from cells transfected as in A, but before fixation cells were unstimulated (−) or stimulated (+) with hypotonic solution (90 mOsm) for 3 min and then fixed, permeabilized, and stained with soluble ASC-RFP (light gray) or nothing (dark gray) and monoclonal α-CARD$_{ASC}$. Data from $n$ = 2–3 independent experiments and quantification of a total of 764 ASC specks. For A–C, and E, data are represented as mean, and error bars represent ± SEM; Mann–Whitney test was used in A and B, ***P < 0.001; Kruskal–Wallis test with Dunn's multiple-comparison post-test was used in C and E, ***P < 0.001; ns, not significant (P > 0.05) difference.

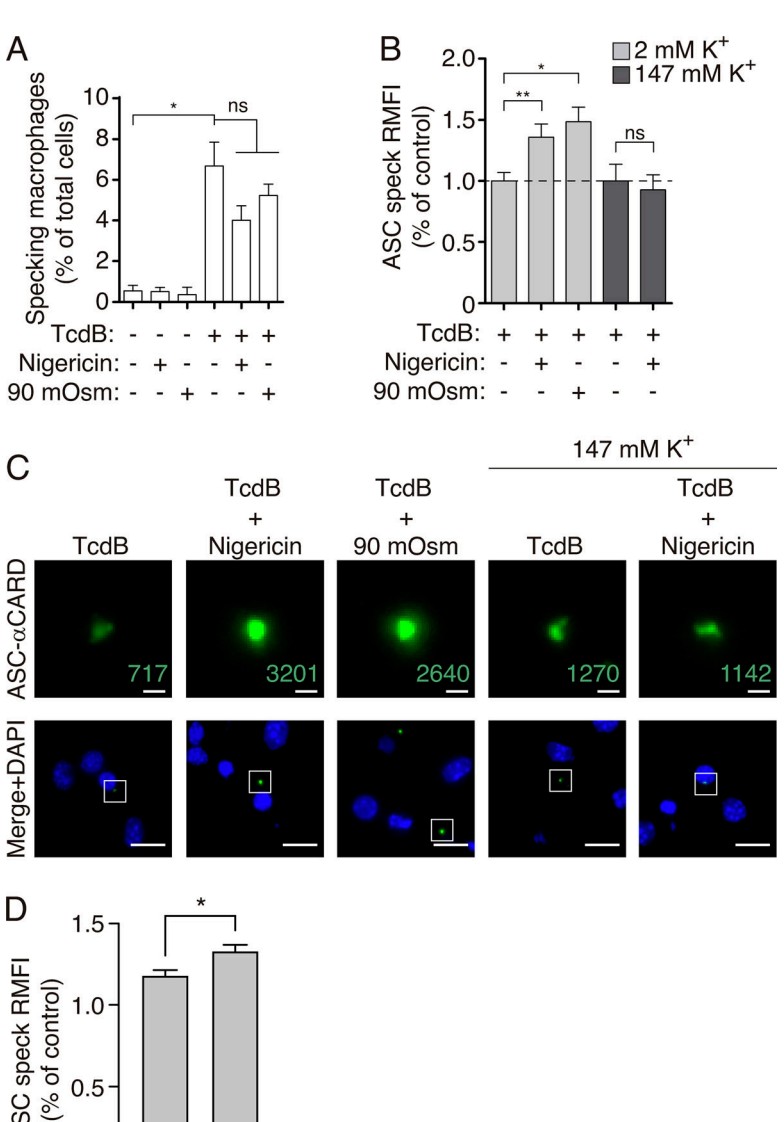

**Figure 6. ASC-speck induced by the Pyrin inflammasome present a structural change in response to K⁺ efflux. (A)** Percentage of ASC specking macrophages derived from *Casp1⁻/⁻* mice stimulated with TcdB (1 µg/ml, 1 h), nigericin (10 µM, 20 min), or hypotonicity (90 mOsm, 20 min), or nigericin or hypotonicity after TcdB treatment as indicated. ASC was stained with monoclonal α-CARD_ASC antibody. Data from *n* = 2 independent experiments and quantification of a total of 1,000 to 2,000 macrophages per treatment. **(B)** Relative mean fluorescence intensity (RMFI) of ASC specks from macrophages treated and stained as in A, but during treatment a buffer with normal extracellular K⁺ (2 mM K⁺, light gray) or high extracellular K⁺ (147 mM K⁺, dark gray) was used. Data from *n* = 2 independent experiments and quantification of a total of 179 ASC specks. **(C)** Representative fluorescent micrographs of macrophages treated and stained as in B. ASC staining with monoclonal α-CARD_ASC antibody is amplified for the selected speck in the top panels (numbers denote RMFI for each of the selected ASC specks) and nuclei were revealed with DAPI (blue); scale bar, top images, 1 µm; bottom images, 10 µm. **(D)** RMFI of ASC specks from *Casp1⁻/⁻* macrophages primed with LPS (1 µg/ml, 2 h) and then treated with PA and FlaA (2:1 ratio, 4 h, Flatox) in the presence of MCC950 (10 µM to avoid NLRP3 activation) and nigericin (10 µM, 20 min). ASC was stained with monoclonal α-CARD_ASC antibody. Data from *n* = 2 independent experiments and quantification of a total of 131 and 159 ASC-specks from Flatox and Flatox + nigericin respectively. For A, B, and D, data are represented as mean, and error bars represent ± SEM; Kruskal–Wallis test with Dunn's multiple-comparison post-test was used in A, *P < 0.05; ns, not significant (P > 0.05) difference; Mann–Whitney test was used in B and D, *P < 0.05; **P < 0.01; ns, not significant (P > 0.05) difference.

the inflammasome (Green et al., 2018). The present study shows that the decrease in K⁺ could also modify the ASC speck structure formed by the decrease in intracellular Cl⁻ to promote the recruitment of the pro-caspase-1^CARD domain, and therefore could also explain caspase-1 activation in conditions where NLRP3 was activated by Cl⁻ efflux.

The recruitment of ASC by NLRP3 occurs via homotypic PYD–PYD interactions (Lu et al., 2014; Cai et al., 2014), and we have previously described how examination of the BRET signals reveals that these interactions are responsive to K⁺ efflux (Compan et al., 2015). Here, we found that the oligomeric structure of ASC mainly formed by PYD–PYD interactions is also responsive to a decrease in intracellular K⁺. This finding is further supported by the fact that the use of a nanobody that specifically blocks ASC CARD–CARD interactions within the ASC speck results in filaments of ASC (Schmidt et al., 2016). These ASC filaments with exposed CARD domains are still able to

undergo a structural change in response to intracellular K⁺ efflux. The ASC^CARD domain is important not only for forming the final ASC speck but also for pro-caspase-1 recruitment and activation (Dick et al., 2016). Therefore, the exposure of ASC^CARD domains induced by K⁺ efflux could be important for modulating caspase-1 activation. Since the ASC^PYD filaments present a rigid core with highly mobile exposed ASC^CARD domains (Sborgi et al., 2015), our data suggest that this flexibility could be responsible for the change observed in the ASC oligomer structure induced after cellular K⁺ efflux. The changes in the ASC-ASC BRET signal suggest overall changes in the quaternary structure of the ASC oligomer as the proximity or orientation of ASC^CARD domains change with respect to each other. However, we cannot rule out that the structural modification in the ASC oligomer observed may be caused by a change within individual ASC molecules in terms of the orientation of their CARD and PYD domains with respect to each other. The formation of ASC^PYD or ASC^CARD

filaments shows that there is no change in the conformation of the individual domains upon oligomerization (Lu et al., 2014; Li et al., 2018). However, these analyses were performed in filaments lacking one of the ASC domains and therefore the orientation of the other domain in the filament is not described, although it is suggested that a flexible linked ASC$^{CARD}$ surrounds the filaments of ASC$^{PYD}$ (Lu et al., 2014; Dick et al., 2016; Sborgi et al., 2015). Furthermore, the structural studies of ASC$^{PYD}$ or ASC$^{CARD}$ filaments were resolved in the absence of K$^+$ (Lu et al., 2014; Li et al., 2018; Sborgi et al., 2015), and our study suggests that the variations in the intracellular ionic environment could be dynamically modulating the structure of the ASC oligomer.

Therefore, strategies that prevent cellular K$^+$ efflux could be adopted to target different pathologies involving inflammasome (Guo et al., 2015), as they would not only target NLRP3 activation but would also affect ASC oligomers (Hornung et al., 2009; Masters et al., 2016; Gao et al., 2016; Zhao et al., 2011). However, we still need further studies to know if the modifications of the ASC oligomer structure could in fact have an impact on inflammasome signaling. In any case, modulating the structure of the ASC oligomer could be useful in treating different pathologies such as Alzheimer's disease, where the ASC oligomer is central to the development of the disease (Venegas et al., 2017). Recently, gout and arthritis have been successfully treated in animal models with specific nanobodies targeting ASC (Bertheloot et al., 2022). Further understanding of how diseases affect ASC activity regulation will help in the development of novel approaches for treating diseases involving inflammasome.

## Materials and methods

### Reagents
*Escherichia coli* LPS O55:B5, DAPI, and nigericin were from Sigma-Aldrich; rabbit polyclonal antibody against caspase-1 p10 (M-20; cat. #sc-514), rabbit polyclonal antibody against ASC (N-15)-R, and horseradish peroxidase-anti-b actin (C4; cat. #sc-22514-R) were from Santa Cruz Biotechnology; mouse monoclonal anti-NLRP3 (Cryo-2; cat. #AG-20B-0014-C100) was from Adipogen; mouse monoclonal anti-ASC (TMS-1) was from BioLegend (cat. #653902); rabbit polyclonal anti-tubulin (cat. #ab4074) from Abcam; rabbit polyclonal anti-renilla luciferase (cat. #PM047) from MBL; ECL horseradish peroxidase conjugated secondary antibody for immunoblot analysis was from GE Healthcare (Anti-Mouse IgG horseradish peroxidase, cat. #NA931; Anti-Rabbit IgG horseradish peroxidase, cat. #NA934); Alexa Fluor-conjugated donkey IgG secondary antibodies (Donkey anti-Mouse IgG AF488, cat. #A-21202; Donkey anti-Rabbit IgG AF488, cat. #R37118; Donkey anti-Mouse IgG AF647, cat. #A-31571; Donkey anti-Rabbit IgG AF647, cat. # A-31573) and ProLong Diamond Antifade Mountant with DAPI were from Life technologies; Fluorescence mounting medium was from DAKO; TcdB was from Enzo.

### Cell culture, treatments, and transfection
Bone marrow was obtained from wild-type C57BL/6, *Casp1*$^{-/-}$*Casp11*$^{-/-}$ mice (Kuida et al., 1995) or *Nlrp3*$^{-/-}$ mice (Mayor et al., 2006) and

differentiated into bone-marrow-derived macrophages (BMDMs) using standard protocols (de Torre-Minguela et al., 2016). All animals were maintained under controlled pathogen-free conditions (20 ± 2°C and a 12-h light–dark cycle), with free access to sterile food and water. HEK293 cells (ATCC CRL-1573) were maintained in DMEM media supplemented with 10% FCS and supplemented with G418 to maintain stable NLRP3-YFP or NLRP3-p.D303N-YFP HEK293 cell lines (Young et al., 2006a; Baroja-Mazo et al., 2014; Tapia-Abellán et al., 2019a, 2021). Cell lines were routinely confirmed to be free of mycoplasma. Lipofectamine 2000 (Life Technologies) was used for the transfection of HEK293 cells as previously described (Young et al., 2006a) using different concentrations of plasmids as stated in the figure legends. It should be noted that non-tagged ASC required a higher concentration of plasmid transfection to induce speck formation (2 µg) whereas C-terminal tagged ASC with YFP or RFP required a lower concentration of plasmid transfection (0.1 µg). ASC expression vectors were generated in pcDNA3.1 (Thermo Fisher Scientific) mammalian expression vector under the cytomegalovirus enhancer–promoter (Martín-Sánchez et al., 2016b). Cells were stimulated for different times with an isotonic solution (300 mOsm) consisting of (in mM) NaCl 147, HEPES 10, glucose 13, CaCl$_2$ 2, MgCl$_2$ 1, and KCl 2; hypotonic solution (90 mOsm) was achieved by diluting the solution 1:4 with distilled sterile water. Alternatively, cells were stimulated with nigericin (10 µM) at different times in an isotonic solution. High extracellular K$^+$ buffer consisted of (in mM) NaCl 2, HEPES 10, glucose 13, CaCl$_2$ 2, MgCl$_2$ 1, and KCl 147.

### ELISA and Western blot
IL-1β release was measured using ELISA kits for mouse IL-1β from R&D following the manufacturer's instructions. ELISA was read in a Synergy Mx (BioTek) plate reader at 450 nm and corrected at 540 and 620 nm. Western blot was carried out as described previously (Compan et al., 2012a; Gillet et al., 2010). Briefly, protein extracts were resolved in 4–12% polyacrylamide gels and electrotransferred. Membranes were probed with different antibodies for caspase-1, NLRP3, ASC, YFP, luciferase, tubulin, or β-actin (antibodies references are in the first section of Material and methods). Membranes were revealed in a ChemiDoc Imaging System (BioRad).

### Bioluminescence resonance energy transfer (BRET) assay
HEK293 cells were co-transfected with a vector encoding for mouse ASC tagged with Luciferase (C-terminus) or YFP (C-terminus; Martín-Sánchez et al., 2016b). After 24 h, transfected cells were seeded on a poly-L-lysine-coated white 96-well plate the day before the assay. The BRET signal was read 5 min after the addition of coelenterazine-h (5 µM; Invitrogen). Luminescence was detected at 37°C in a Synergy Mx plate reader (Biotek) using two filters for emission at 485 ± 20 and 528 ± 20 nm. The BRET ratio was calculated as the difference between the 528 and 485 nm emission ratio of R-Luc-ASC and YFP-ASC divided by the difference between the 528 and 485 nm emission ratio of the R-Luc-ASC protein alone. Results are expressed in milliBRET (mBRET) units normalized to basal signal as previously described (Martín-Sánchez et al., 2016a).

## Immunofluorescence

HEK293 cells or macrophages were seeded on poly-L-lysine coverslips 24 h before use. After transfection and/or stimulation, cells were fixed with 4% formaldehyde, blocked using 2% bovine serum albumin (Sigma-Aldrich), and permeabilized with 0.1% Triton X-100 (Sigma-Aldrich). ASC was stained using a primary antibody anti-ASC (polyclonal anti-PYD ASC, Santa Cruz or monoclonal anti-CARD ASC, 1:1,000 dilution; BioLegend) or anti-Luciferase (Abcam; 1:500 dilution) and an Alexa Fluor AF488 or AF647 conjugated secondary antibody (1:200 dilution; Life Technologies). ProLong Diamond Antifade Mountant with DAPI was used as a mounting medium. Images were acquired at room temperature with a Nikon Eclipse Ti microscope equipped with a 20× S Plan Fluor objective (numerical aperture 0.45), a 40×S Plan Fluor objective (numerical aperture 0.6), and a 60×S Plan Apo Vc objective (numerical aperture 1.40), and a digital Sight DS-QiMc camera (Nikon) with a Z optical spacing of 0.2 μm and 387 nm/447 nm, 482 nm/536 nm, 543 nm/593 nm, and 650 nm/668 nm filter sets (Semrock) and NIS Elements software (Nikon). Images were processed using ImageJ software (National Institutes of Health), and the maximum-intensity projection images are shown in the results.

## Isolation of soluble caspase-1 CARD-EGFP and ASC-RFP

HEK293 cells transfected with a vector encoding mouse caspase-1 CARD domain tagged with EGFP (0.2 μg) or ASC tagged with RFP (0.2 μg) were lysed in CHAPS (20 mM HEPES-KOH, pH 7.5, 5 mM MgCl2, 0.5 mM EGTA, 0.1 mM PMSF and 0.1% CHAPS) by passing through a syringe on ice (25-gauge needle, 20 times), and the supernatant containing the soluble expressed proteins was obtained by sequential centrifugation at 4°C. The fluorescence in the supernatants was checked using a plate reader and the absence of oligomers was confirmed by fluorescence microscopy.

## Soluble caspase-1 CARD-EGFP and ASC-RFP recruitment assay

Poly-L-lysine coverslip seeded HEK293 cells were transfected with mouse ASC (2 μg) 24 h before use. Cells were fixed with 4% formaldehyde after stimulation with nigericin (10 μM) or hypotonic solution (90 mOsm), then washed with PBS, blocked using 2% bovine serum albumin (Sigma-Aldrich), and permeabilized with 0.1% Triton X-100 (Sigma-Aldrich). After that, cells were incubated with soluble caspase-1 CARD domain-EGFP (Jenster et al., 2023; 1:50 dilution in PBS) or ASC-RFP (1:5 dilution in PBS) for 30 min and then fixed for 10 min with 4% formaldehyde, washed with PBS, and stained for ASC using a primary antibody against the CARD domain (1:1,000 dilution; BioLegend) followed by an Alexa-Fluor-conjugated-secondary antibody (1:200 dilution; Life Technologies). Nuclei were stained using DAPI (1 μg/ml; 10 min), and the coverslip was mounted with a fluorescence-mounting medium.

## Measurement of intracellular K+

12-well plates with $10^6$ HEK293/well were stimulated for 1 h with hypotonic solution (90 mOsm) or for 30 min with nigericin (10 μM) at 37°C, then cells were briefly and quickly washed with nuclease-free water, and then immediately after cells were scraped in 200 μl/well of nuclease-free water followed by three freeze–thaw cycles. Lysates were centrifuged at 16,000 × g for 10 min at 4°C, and the supernatants were stored at –80°C until K+ concentration was quantified by indirect potentiometry using a Cobas 6000 with ISE module (Roche).

## Intensity fluorescence measurement

The fluorescence intensity of the ASC oligomers was analyzed using ImageJ software. For this, a region of interest (ROI) was drawn around the oligomer, measuring the average pixel intensity within each selected area and subtracting an average background value from three selected ROIs within the same image. The fluorescence intensity values obtained were normalized relative to the control inside the same experiment.

## Statistical methods

The data are presented as the mean ± SEM. Data were analyzed using Prism (GraphPad) software using non-parametric tests if data was not following a normal distribution according to D'Agostino and Pearson normality test ($n > 5$) or the Shapiro–Wilk test ($n < 5$). Non-parametric two groups comparison was done using the Mann–Whitney test (two tails, 95% confidence level), and the Kruskal–Wallis test with Dunn's multiple-comparison post-test was used to determine the differences among more than two groups. Parametric two groups comparison was done using unpaired $t$ test (two tails, 95% confidence level). ***P < 0.001; **P < 0.01; *P < 0.05; ns, not significant (P > 0.05) difference.

## Online supplemental material

Fig. S1 shows that K+ efflux induces NLRP3 inflammasome activation. Fig. S2 shows that ASC speck conformational change detected by BRET is dependent on K+ efflux. Fig. S3 shows that the ASC speck could be differentially labeled with antibodies. Fig. S4 shows that K+ efflux does not control Pyrin inflammasome activation. Table S1 shows the differential detection of soluble vs. oligomeric ASC with different antibodies.

## Data availability

The data are available from https://doi.org/10.5061/dryad.n8pk0p31b.

## Acknowledgments

The authors thank Dr. Isabelle Couillin (Experimental and Molecular Immunology and Neurogenetics, University of Orleans, Orleans, France) for caspase-1-deficient mice and the SPF-animal house from IMIB-Arrixaca for mice colony maintenance. We thank the staff at Dr. Pelegrin's laboratory for their comments and suggestions.

This work was supported by grants to P. Pelegrin as principal investigator from MCIN/AEI/10.13039/501100011033 (grant PID2020-116709RB-I00), the Ministerio de Economia, Industria y Competitividad–Fondo Europeo de Desarrollo Regional (project no. SAF2017-88276-R), Fundación Séneca (grants 20859/PI/18 and 21897/PI/22), the European Research Council (grants ERC-2013-CoG project no. 614578 and ERC-2019-PoC project no. 899636), the Instituto Salud Carlos III (grant DTS21/00080), and

the EU Horizon 2020 project PlasticHeal (grant 965196). F. Martín-Sánchez was supported by Sara Borrell postdoc grant from the Instituto Salud Carlos III (CD12/00523). F.I. Schmidtis funded by the Deutsche Forschungsgemeinschaft (DFG, German Research Foundation) through the Emmy Noether Programme (SCHM 3336-1-1) and Germany's Excellence Strategy–EXC2151–390873048. The authors would like to acknowledge networking support by COST Action BM-1406 and CA21130 via WP meeting interaction.

Author contributions: F. Martín-Sánchez, V. Compan, A. Peñín-Franch, A. Tapia-Abellán, M.C. Baños, and A.I. Gómez performed the experiments; F.I. Schmidt provided nanobodies, Flatox, and caspase-1^CARD constructions, and interpreted data; F. Martín-Sánchez and P. Pelegrin analyzed and interpreted the data, and prepared the figures; F. Martín-Sánchez, V. Compan, and F.I. Schmidt helped with final manuscript preparation; P. Pelegrin conceived the experiments, wrote the paper, provided funding, and helped in the conception and overall supervision of this study.

Disclosures: All authors have completed and submitted the ICMJE Form for Disclosure of Potential Conflicts of Interest. F. Schmidt reported personal fees from Odyssey Therapeutics and other from Odyssey Therapeutics outside the submitted work. P. Pelegrin reported "He is an inventor in a patent filed on March 2020 by the Fundación para la Formación e Investigación Sanitaria de la Región de Murcia (PCT/EP2020/056729) for a method to identify NLRP3-immunocompromised sepsis patients. He is a co-founder of Viva In Vitro Diagnostics SL, but declares that the research was conducted in the absence of any commercial or financial relationships that could be construed as a potential conflict of interest." No other disclosures were reported.

Submitted: 9 March 2020

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

# Supplemental material

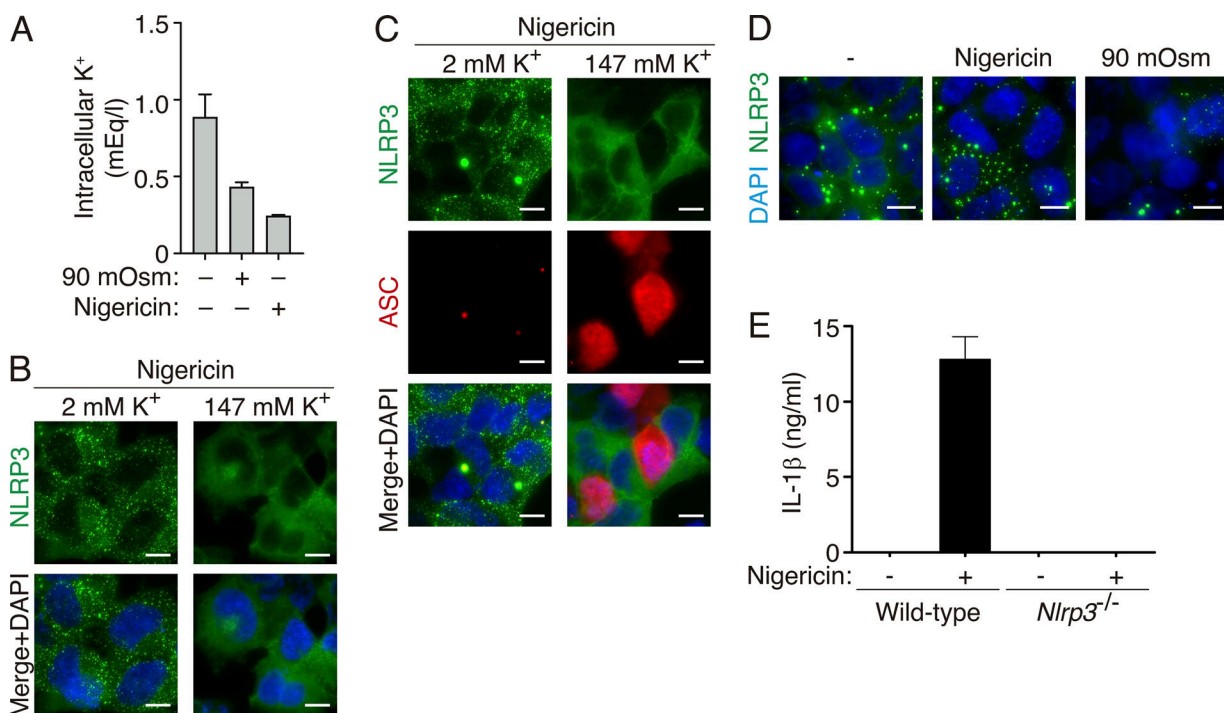

Figure S1.   **K⁺ efflux induces NLRP3 oligomerization. (A)** Intracellular K⁺ concentration of HEK293 cells exposed for 1 h to hypotonic buffer (90 mOsm) or stimulated for 30 min with nigericin (10 µM) (n= 5–8 biological replicates). **(B)** Representative fluorescent micrographs of NLRP3-YFP HEK293 stable cell line and treated with nigericin (10 µM, 30 min) in normal extracellular K⁺ (2 mM K⁺) or high extracellular K⁺ (147 mM K⁺) as indicated. NLRP3-YFP (green) and nuclei were revealed with DAPI (blue); bar 10 µm. **(C)** Representative fluorescent micrographs of NLRP3-YFP HEK293 stable cell line transfected with a plasmid encoding for ASC (1 µg) and treated as in A. NLRP3-YFP (green), ASC was stained with polyclonal α-PYD$_{ASC}$ antibody (AF647, red), and nuclei were revealed with DAPI (blue); bar 10 µm. **(D)** Representative fluorescent micrographs of NLRP3-YFP D303N HEK293 stable cell line unstimulated (–) or stimulated with nigericin (10 µM, 30 min) or hypotonic solution (90 mOsm, 1 h) as indicated. NLRP3-YFP D303N (green), and nuclei were revealed with DAPI (blue); bar 10 µm. **(E)** ELISA for IL-1β in bone-marrow-derived macrophage supernatants after LPS (1 µg/ml, 4 h) followed by nigericin (10 µM, 30 min) stimulation (n = 4–6 biological replicates). For A and E data are represented as mean and error bars represent ± SEM.

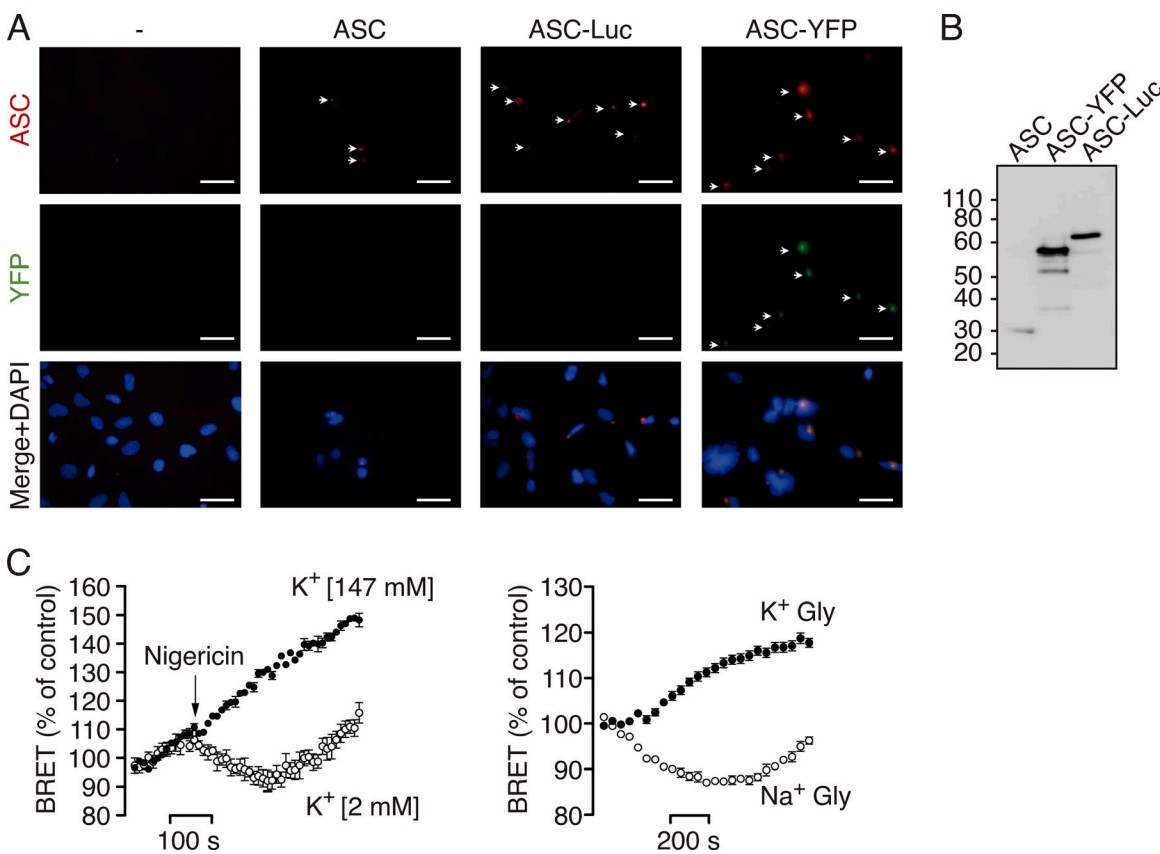

Figure S2.   **ASC speck conformational change detected by BRET is dependent on K⁺ efflux. (A)** Representative fluorescent micrographs of HEK293 cells transfected with a plasmid encoding for untagged ASC, ASC-YFP, or ASC-Luc. ASC was stained with polyclonal α-PYD$_{ASC}$ antibody (AF647, red), ASC-YFP (green), and nuclei were revealed with DAPI (blue); scale bar, 20 μm. **(B)** Immunoblot for ASC of cells using the polyclonal α-PYD$_{ASC}$ antibody transfected as in A. **(C)** Kinetic of net BRET signal in HEK293 cells transfected with the BRET donor ASC-Luc and the acceptor ASC-YFP in response to nigericin (left) or cell swelling induced by glycine solution (Na⁺-Gly, right). Nigericin was applied in normal extracellular K⁺ (2 mM K⁺) or high extracellular K⁺ (147 mM K⁺) buffer as indicated. Cell swelling was performed in sodium- (Na⁺-Gly) or potassium (K⁺-Gly)-based buffer as indicated. Data are represented as mean and error bars represent ± SEM ($n$ = 3–8 biological replicates).

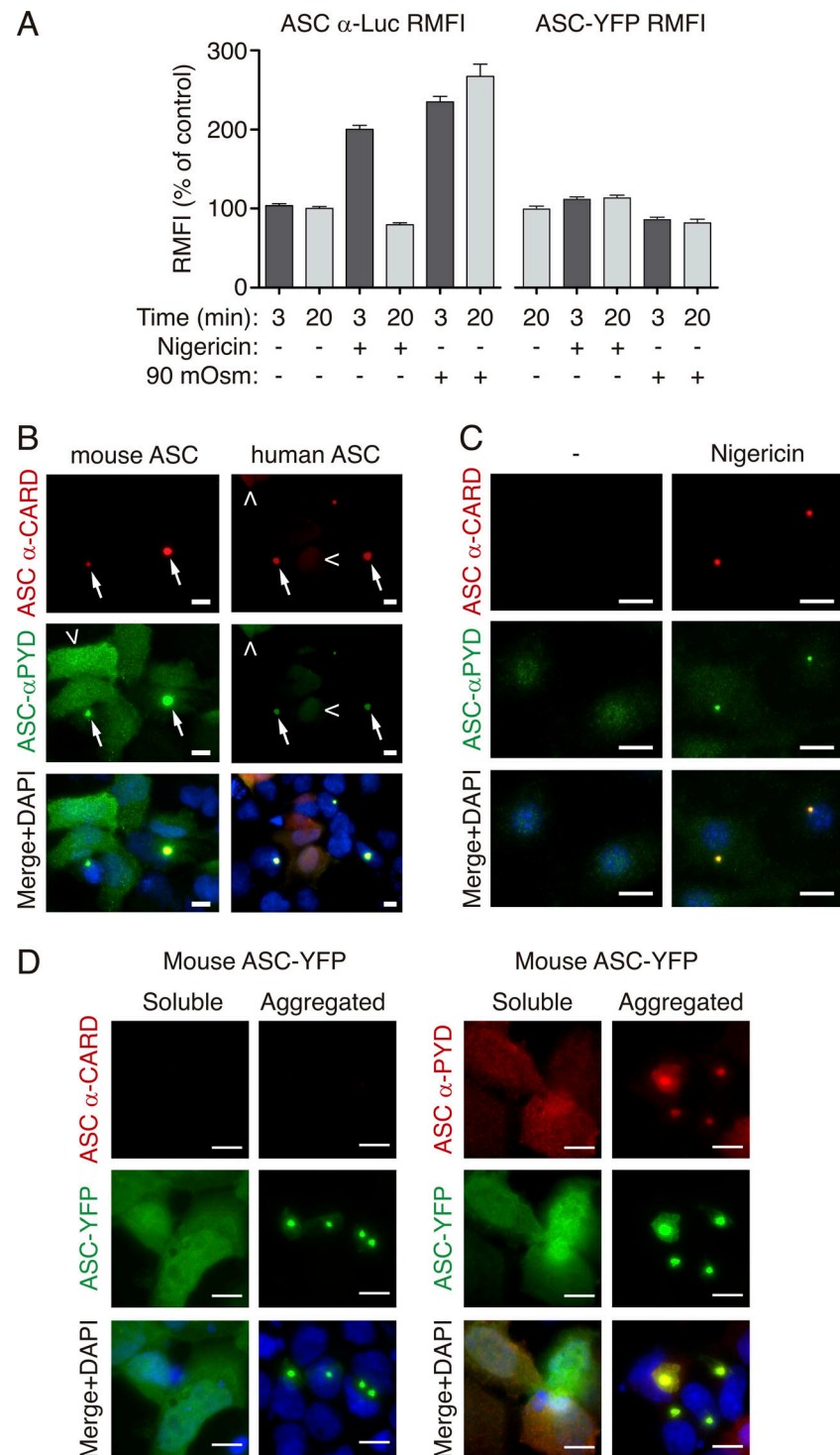

Figure S3. **ASC speck is differentially labeled with antibodies. (A)** α-Luc antibody (left) or YFP (right) fluorescence of HEK293 cells transfected with plasmids encoding for ASC-YFP (0.2 µg) and ASC-Luc (0.1 µg), unstimulated (control) or stimulated with nigericin (10 µM) or hypotonic solution (90 mOsm) for the indicated times. These data correspond to Fig. 3, A and B. Data are represented as mean, and error bars represent ± SEM (ASC α-Luc RFMI data is from $n = 3$ independent experiments and quantification of a total of 2,084 ASC-specks; ASC-YFP RFMI data is from $n = 1$–2 independent experiments and quantification of a total of 2,211 ASC-specks). **(B)** Representative fluorescent micrographs of HEK293 cells transfected with a plasmid encoding for untagged mouse ASC (2 µg, left) or untagged human ASC (1 µg, right). ASC was co-stained with monoclonal α-CARD$_{ASC}$ antibody (AF647, red) or with polyclonal α-PYD$_{ASC}$ antibody (AF488, green), and nuclei were revealed with DAPI (blue); scale bar, 10 µm. **(C)** Representative fluorescent micrographs of mouse macrophages primed with LPS (1 µg/ ml, 4 h) and then stimulated with nigericin (10 µM, 10 min). ASC was stained with monoclonal α-CARD$_{ASC}$ antibody (AF647, red) or with polyclonal α-PYD$_{ASC}$ antibody (AF488, green), and nuclei were revealed with DAPI (blue); scale bar, 10 µm. **(D)** Representative fluorescent micrographs of HEK293 cells transfected with a plasmid encoding for ASC-YFP (0.2 µg). ASC-YFP was staining with monoclonal α-CARD$_{ASC}$ antibody (AF647, red, left panels) or with polyclonal α-PYD$_{ASC}$ antibody (AF647, red, right panels), ASC-YFP (green) and nuclei was revealed with DAPI (blue); scale bar, 10 µm.

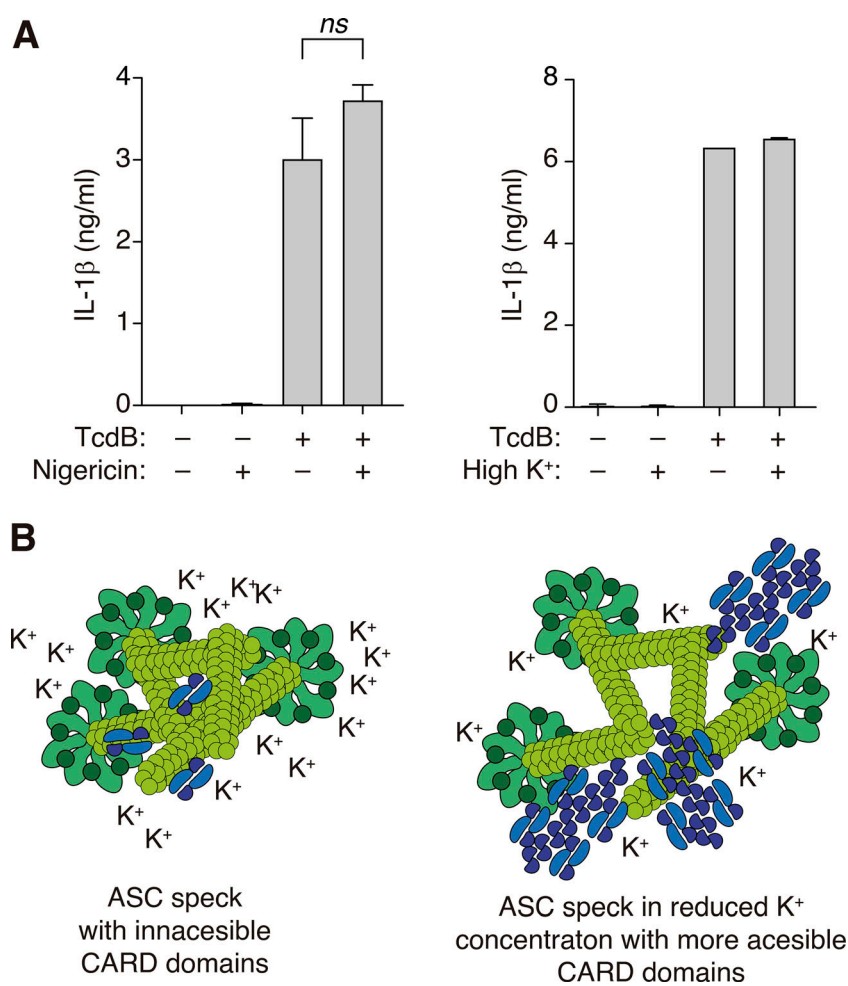

**A**

**B**

ASC speck
with innacesible
CARD domains

ASC speck in reduced K⁺
concentraton with more acesible
CARD domains

Figure S4. **K⁺ efflux do not control Pyrin inflammasome activation. (A)** IL-1β release from bone-marrow-derived macrophages from *Nlrp3⁻ᐟ* mice stimulated for 4 h with LPS (1 µg/ml) and then for 1 h with TcdB (1 µg/ml), and during the last 30 min treated with nigericin (25 µM, left) or TcdB was applicated in a buffer with high K⁺ (147 mM K⁺, right). Data are represented as mean, and error bars represent ± SEM (*n* = 2–7 independent experiments). **(B)** Proposed model of the regulation of the ASC speck by K⁺ efflux.

Provided online is Table S1. Table S1 shows the detection of ASC with α-PYD$_{ASC}$ (SantaCruz) and α-CARD$_{ASC}$ (BioLegend) antibodies used in this study.

