## [Peer Review File · The Journal of Cell Biology]

ASC oligomer favors caspase-1 CARD domain recruitment after intracellular potassium efflux

Fátima Martín-Sánchez, Vincent Compan, Alejandro Peñín-Franch, Ana Tapia-Abellán, Ana I Gómez-Sánchez, Maria Baños, Florian Schmidt, and Pablo Pelegrin

Corresponding Author(s): Pablo Pelegrin, Instituto Murciano de Investigación Biosanitaria

Review Timeline:

Submission Date:	2020-03-09
Editorial Decision:	2020-06-10
Revision Received:	2023-02-21
Editorial Decision:	2023-04-03
Revision Received:	2023-04-21

Monitoring Editor: Craig Roy

Scientific Editor: Lucia Morgado-Palacin

Transaction Report:

DOI: <https://doi.org/10.1083/jcb.202003053>

June 10, 2020

Re: JCB manuscript #202003053

Dr. Pablo Pelegrin
Biomedical Research Institute of Murcia
Carretera Buenavista
Murcia 30120
Spain

Dear Dr. Pelegrin,

Thank you for submitting your manuscript entitled "ASC oligomer favor caspase-1CARD domain recruitment after intracellular potassium efflux". The manuscript was assessed by expert reviewers, whose comments are appended to this letter. We invite you to submit a revision if you can address the reviewers' key concerns, as outlined here.

All three reviewers feel this work provides novel insight on ASC function in response to low potassium but that several conclusions require additional verification. The demonstration that low potassium can result in ASC-mediated activation of caspase-1 and cleavage of Gasdermin D independent of NLR function would address several concerns.

GENERAL GUIDELINES:

Text limits: Character count for an Article is < 40,000, not including spaces. Count includes title page, abstract, introduction, results, discussion, acknowledgments, and figure legends. Count does not include materials and methods, references, tables, or supplemental legends.

Figures: Articles may have up to 10 main text figures. Figures must be prepared according to the policies outlined in our Instructions to Authors, under Data Presentation, <http://jcb.rupress.org/site/misc/ifora.xhtml>. All figures in accepted manuscripts will be screened prior to publication.

Supplemental information: There are strict limits on the allowable amount of supplemental data. Articles may have up to 5 supplemental figures. Up to 10 supplemental videos or flash animations are allowed. A summary of all supplemental material should appear at the end of the Materials and methods section.

As you may know, the typical timeframe for revisions is three to four months. However, we at JCB realize that the implementation of social distancing and shelter in place measures that limit spread of COVID-19 also pose challenges to scientific researchers. Lab closures especially are preventing scientists from conducting experiments to further their research. Therefore, JCB has waived the revision time limit. We recommend that you reach out to the editors once your lab has reopened to decide on an appropriate time frame for resubmission. Please note that papers are generally considered through only one revision cycle, so any revised manuscript will likely be either accepted or rejected.

Thank you for this interesting contribution to Journal of Cell Biology. You can contact us at the journal office with any questions, cellbio@rockefeller.edu or call (212) 327-8588.

Sincerely,

Craig Roy, Ph.D.
Monitoring Editor

Reviewer #1 (Comments to the Authors (Required)):

The manuscript by Sánchez and colleagues show that the ASC specks could change its structure under conditions of low intracellular potassium, which allows the CARD domain of ASC to be more accessible for the recruitment of the CARD domain of pro-caspase-1. To demonstrate this, the authors performed a series of nice cell-biology imaging analyses using different antibody-labeling strategies. This is the first time to suggest that the ASC-containing inflammasome pathways can sense low potassium concentration independently of the NLRP3 sensor. This finding may change our understanding of the regulation of other inflammasome pathways mediated by NLRP1, Pyrin and NLRC4. They further propose an interesting idea that upon inflammasome activation the initial stage of low-level gasdermin D pore formation may decrease intracellular potassium concentration, which could positively amply the extent of inflammasome activation. Below are a few comments that authors should consider to address before publication of the story.

1. Why did ASC form specks in the control group of Fig. 1H as only 0.1 ug of ASC-RFP was transfected into the HEK293T cells? In fact, the entire Fig. 1 probably can be moved to the supplementary part as they are all confirming previous observations.
2. In Fig 2, the authors showed that ASC specks undergo a structural change following nigericin or hypotonic stimulation, which was achieved by using a BRET assay. A bit of introduction of the BRET assay is needed in the text description so that the data can be easily followed. As the conclusion is solely derived from the BRET assay with transfected ASC constructs, the reviewer feels that more extensive analyses under a physiological context are needed to support the conclusion.
3. Why was staining of ASC specks by the Luc antibodies more intense after the treatment, while ASC-YFP fluorescence remained constant whatever the treatment? A better control for this experiment is to compare the YFP fluorescence intensity with that of staining with an anti-YFP antibody. In Fig. 3E, why RMFI of human ASC increased upon hypotonic stimulation in 3 min but decreased markedly after 20 min, which is inconsistent with the data in Fig. 3B? Also, there seems to be quite a big difference between mouse and human ASC when comparing the Fig. 3D and 3E data. What is the explanation for this discrepancy?
4. In Fig. 6, TcdB-induced ASC oligomerization was not that obvious. In fact, TcdB is a highly potent trigger of the pyrin inflammasome and can robustly induce ASC foci formation. Is there a technical issue here? Besides, the authors should investigate whether the structural change of ASC specks also participates in other inflammasome pathways.
5. The authors showed that nigericin and hypotonic stimulation were not able to alter the number of endogenous ASC specks in macrophages upon TcdB stimulation. Does this suggest that the structural change of ASC specks after nigericin or hypotonic stimulation only occur in the artificial cell system and is dispensable for physiological caspase-1 activation?
6. The authors propose that the initial low-level gasdermin D pore formation may positively amply the extent of inflammasome activation via the decrease of intracellular potassium concentration. This is an interesting idea of potential physiological significance. Can the authors provide experimental evidences supporting this?
7. The language of the presentation needs to be significantly proved as there are many grammatic errors and awkward sentences.

Reviewer #2 (Comments to the Authors (Required)):

Comments on the study « ASC oligomer favor caspase-1CARD domain recruitment after intracellular potassium efflux » from Martin-Sanchez et al.,

In their study the authors adress the importance of intracellular potassium (K⁺) concentration on ASC CARD domain accessiility to Caspase-1. These findings are worth investigating further, as so far, potassium has only been described as a central signal for the Nlrp3 inflammasome activation and assembly, but not for ASC-Caspase-1 interaction. The study is conducted with very interesting tools to detect intraASC oligomer structure. Yet, some parts seem to be a bit overstated in absence of proper experiments.

Point 1/

One puzzling thing to me is that the authors suggest that intracellular potassium is required for better Casp-1 recruitment to the oligomerized ASC in response to Pyrin+potassium efflux triggering condition. Could the authors test whether inhibiting

potassium efflux in their conditions alter the ASC speck intensity ? But also, is potassium efflux altering the ASC-Casp-1 interaction after a single PYRIN inflammasome stimulation (meaning without any further stimulation with nigericin or hypotonic media)?

Point 2/

But also, are the author findings also applying to the three other inflammasomes, namely NLRC4, CARD8 and NLRP1, where the ASC oligomerization status is different given they all express a CARD domain? One could speculate that intracellular [K+] might also regulate CARD-containing sensors and their direct, ASC-independent, Caspase-1 recruitment ? Could the authors test this ?

Point 3/

A strong missing point is the lack of translation of the authors findings in inflammasome function. Meaning, could the author use K+ High [] or K+ low [] to translate their findings in Caspase-1 activity ? Gasdermin D cleavage ? IL1beta maturation and release ? Cell death ?

Reviewer #3 (Comments to the Authors (Required)):

Martin-Sanchez et al present an intriguing study on the role of intracellular potassium ion concentration in regulating the structure of the key inflammasome adapter protein ASC. Previous studies have shown that K+ efflux is essential for NLRP3 activation in response to almost all NLRP3 stimuli (except R837 and alternative NLRP3 activation in monocytes). The authors largely use a HEK cell overexpression system as their model (as they have in numerous previous studies). Using this system, they show that ASC can form functional specks even in the absence of an inflammasome sensor. Using a BRET biosensor approach, they then show that ASC undergoes a structural change when HEK cells are stimulated with nigericin or hypotonic medium (which are known to cause K+ efflux). They support their BRET observations with an antibody-based approach and show that in similar conditions (HEK cells stimulated with nigericin or hypotonic medium) there is a change in the binding ability of an antibody directed to the CARD domain of ASC - indicating a structural change in ASC. The authors then suggest that this structural change in ASC enhances caspase-1 binding to ASC (CARD-CARD interaction). In macrophages where the pyrin inflammasome has been activated, they can observe the change in ASC-CARD antibody binding when cells are also treated with nigericin or hypotonic medium. Overall the authors conclude that potassium efflux not only regulates NLRP3 but also the structure of the ASC speck and thus will influence signalling from other inflammasomes such as pyrin and AIM2.

While the data presented here are sound, and I believe that this is potentially an important and novel observation about the cell biology of inflammasomes, I think that more experiments are needed to support the conclusions drawn by the authors. My major and minor comments are listed below but for clarity the two key questions that need to be resolved/addressed are:

1. Whether there is any role for other ions such as chloride in this phenotype.
2. Whether the increased availability of the CARD domain has any influence on caspase-1 activation particularly in an immune cell such as a macrophage.

Major Comments

General

1. A diagram that shows how the authors think the ASC structure changes would be very helpful.

Introduction

2. The introduction is missing some key references (it is not sufficient to just cite reviews 1 and 2) which must be included and discussed to provide the correct scientific context for this study (the concept of ion flux regulating caspase-1 activation has been around for decades). Changes in intracellular potassium were first linked to IL-1beta processing by Perregaux and Gabel (PMID: 8195155. J Biol Chem. 1994 May 27;269(21):15195-203. Interleukin-1 beta maturation and release in response to ATP and nigericin. Evidence that potassium depletion mediated by these agents is a necessary and common feature of their activity.) The activation of the NLRP3 inflammasome by low K+ was first reported by Petrilli et al (Pétrilli V, Papin S, Dostert C, Mayor A, Martinon F, Tschopp J. Cell Death Differ. 2007 Sep;14(9):1583-9. Epub 2007 Jun 29. Activation of the NALP3 inflammasome is triggered by low intracellular potassium concentration. PMID: 17599094)

Results

3. Figure 1A - K+ has not been measured here. This could be attributed to cell swelling or changes in other ions such as Cl-. The changes in intracellular K+ levels (that are assumed to occur) should actually be demonstrated in this HEK system.
4. Fig S1A-B (and Fig. 2E, S2C) - Although a high K+ buffer blocks these effects the specific details of this buffer haven't been provided in the methods. I would presume this contains a high concentration of KCl so could the effect be due to high Cl-? High extracellular Cl- has been shown to inhibit NLRP3 activation (Tang et al 2017 PMID: 28779175 and Domingo-Fernandez et al 2014 PMID: 28576828).
5. Fig 1E This assay shows that caspase-1 autocatalytic cleavage is occurring. It would be more convincing if an additional

assay was included to demonstrate caspase-1 activity such as staining with FAM-YVAD-FMK or by measuring substrate processing (e.g. co-transfection of a pro-IL-1beta expressing plasmid and measure IL-1beta cleavage).

6. Fig 1G and Fig 1H are the filaments we see in these two images supposed to be similar or different? The high-magnification image of the ASC structure in Fig 1G is very compelling. Could this be used to analyse the same structures shown in Fig 1H with ASC-RFP?

7. Fig S2A, Fig 2A, Fig 3A: ASC-luciferase specks appear to be bigger and on the outside of the co-expressed specks while ASC-YFP is denser more concentrated. Why might this be?

8. Figure 3A,B - ASC-YFP intensity should also be quantified here as it looks like there is increased fluorescence after 20 min nigericin. Is it possible that the nigericin and hypotonic treatments non-specifically affect YFP and/or luciferase? It might be useful to include a control where a different protein (i.e. not ASC) is tagged with luciferase and/or RFP to exclude this possibility.

9. Figure 3 'These data suggest that upon K⁺ efflux, the ASCCARD domain is more accessible to antibody staining'. I think this is an overinterpretation of these results. For example, the antibody staining could be enhanced because of the plasma membrane changes caused by nigericin treatment.

10. Fig. 4A, S3C - I'm not sure it's correct that this antibody can only detect oligomerised ASC. For example, in PMID: 27221487 Fig 3B there is clear staining non-oligomerised ASC using the Biolegend antibody (HASC-71) - is this difference potentially due to antibody concentration?

11. Figure 6 - TcdB stimulation can also activate other signalling pathways (e.g. MAPK - see review PMID: 27108093) so there is the potential that it could prime NLRP3. Another inflammasome should be tested - I would suggest AIM2 activation which does not require priming.

12. Figure 6B-C the percentage change in ASC-CARD staining is very small here and doesn't seem to be consistent with the representative images shown in 6C (i.e 1-1.4% vs 707-3202). This should be presented alongside data from cells where NLRP3 has been activated for comparison.

13. Figure 6B-C The authors need to test if this observation results in enhanced caspase-1 activation. For example, by quantifying caspase-1 activity using a fluorescent substrate such as FAM-YVAD-FMK or by measuring caspase-1 substrate cleavage such as IL-1beta, IL-18 or gasdermin-D by ELISA or Western blotting.

Minor Comments

General

1. Some editing of the grammar would enhance the clarity of the manuscript.

Introduction

2. Line 2/3: 'NLRP3 is a multiprotein complex, whose oligomerization occurs' suggest change to 'The NLRP3 inflammasome is a multiprotein complex whose activation occurs...' or 'NLRP3 oligomerises...' Just to be clear that NLRP3 itself is not a multiprotein complex.

Results

3. Figure S1C - Based on these images there does look like there is less D303N oligomerisation in the 90 mOsm treatment. Perhaps this could be quantified or other representative images shown.

4. Western blots should have molecular weight markers added.

5. Some of the scale bars on the microscopy images are not that visible - could this be made clearer with a thicker or larger line?

6. Fig S2A is missing scale bars.

7. Fig 2G - a legend showing what the block dots and grey dots are would be really helpful for the reader here.

8. Figures 3 C-E - I think it would be useful to show some representative images here. They could be included as a supplemental figure.

9. Figure S3B - the representative images are not very clear. If possible, a higher magnification i.e. the same as the magnification in S3A should be presented.

10. Figure 4B and S3D-E are these data from the same experiments? It would be clearer to show the data from the high K⁺ on the same graph.

11. Figure 5A Why is there so much less CARD-EGFP staining? Why was ten-fold less protein used?
12. Figure 5C What is the control condition here? Is it untreated cells (as shown in 5A)?

Pablo Pelegrín, PhD
Scientific Deputy Director & Principal Investigator
Biomedical Research Institute of Murcia
University of Murcia
30120 Murcia (Spain)

February 21, 2023

Dear Dr Roy and O'Donnell, Editors of JCB,

With this letter we are resubmitted a revised version of our manuscript entitled '**ASC oligomer favor caspase-1CARD domain recruitment after intracellular potassium efflux**' (JCB manuscript #202003053) for consideration for publication in JCB.

First of all, we would like to apologize since we had several complications related with the revision of the above-mentioned manuscript and we are aware that even we had a waived revision time limit, we might be out of revision, and we leave at your discretion to continue peer-review process at JCB. After COVID lock-down, the main postdoc of the paper (and first author) went to other lab, and with the initial restrictions I had nobody able to handle the required experiments. During the last year we have been advancing in the experiments with a trained PhD student of the lab, but he was unable to devote too much time to new experiments as he was mainly focused on his PhD project. Therefore, the advance in obtaining new data for the revision has not been as fast as we wanted. Now, we are at disposition to submit a revised version of this study with enough strength to answer all the reviewers' points (see below the point by point rebuttal letter).

In this version of our manuscript, we have revised the text and figures in accordance to the reviewers' suggestions, and all the comments have been addressed in the point-by-point rebuttal letter. Changes in the manuscript are shown in word tracked tool in the revised submission.

We hope our additions and modifications would satisfy all the points raised by the reviewers, we consider that their constructive comments have greatly improved the final quality of the manuscript.

We are sorry for the delay in getting a revised manuscript.

Thank you very much for your consideration.
Yours faithfully,

Pablo Pelegrín on behalf of all authors.

Point by point rebuttal letter:

JCB manuscript #202003053

Manuscript entitled "ASC oligomer favor caspase-1CARD domain recruitment after intracellular potassium efflux".

General:

All three reviewers feel this work provides novel insight on ASC function in response to low potassium but that several conclusions require additional verification. The demonstration that low potassium can result in **ASC-mediated activation of caspase-1 and cleavage of Gasdermin D independent of NLR** function would address several concerns.

Editorial points:

Text limits: Character count for an Article is < 40,000, not including spaces. Count includes title page, abstract, introduction, results, discussion, acknowledgments, and figure legends. Count does not include materials and methods, references, tables, or supplemental legends.

Answer: Character count of the revised manuscript is 31,076, not including spaces.

Figures: Articles may have up to 10 main text figures.

Answer: The manuscript presents 6 Figures.

Supplemental information: There are strict limits on the allowable amount of supplemental data. Articles may have up to 5 supplemental figures. Up to 10 supplemental videos or flash animations are allowed. A summary of all supplemental material should appear at the end of the Materials and methods section.

Answer: Supplemental information has 4 Figures and 1 Table.

Reviewer #1 (Comments to the Authors (Required)):

The manuscript by Sánchez and colleagues show that the ASC specks could change its structure under conditions of low intracellular potassium, which allows the CARD domain of ASC to be more accessible for the recruitment of the CARD domain of pro-caspase-1. To demonstrate this, the authors performed a series of nice cell-biology imaging analyses using different antibody-labeling strategies. This is the first time to suggest that the ASC-containing inflammasome pathways can sense low potassium concentration independently of the NLRP3 sensor. This finding may change our understanding of the regulation of other inflammasome pathways mediated by NLRP1, Pyrin and NLRC4. They further propose an interesting idea that upon inflammasome activation the initial stage of low-level gasdermin D pore formation may decrease intracellular potassium concentration, which could positively amplify the extent of inflammasome activation. Below are a few comments that authors should consider to address before publication of the story.

1. Why did ASC form specks in the control group of Fig. 1H as only 0.1 ug of ASC-RFP was transfected into the HEK293T cells? In fact, the entire Fig. 1 probably can be moved to the supplementary part as they are all confirming previous observations.

Answer: The ASC tagged with RFP or YFP at C-terminal has a trend to oligomerize easily than non-tagged ASC, therefore transfecting 0.1ug ASC-RFP was enough to induce speck formation. A brief sentence explaining this has been added to the Methods section (page 15-16):

“It should be noted that that non-tagged ASC required a higher concentration of plasmid transfection to induce speck formation (2 ug) whereas C-terminal tagged ASC with YFP or RFP required a lower concentration of plasmid transfection (0.1 ug)”

2. In Fig 2, the authors showed that ASC specks undergo a structural change following nigericin or hypotonic stimulation, which was achieved by using a BRET assay. A bit of introduction of the BRET assay is needed in the text description so that the data can be easily followed. As the conclusion is solely derived from the BRET assay with transfected ASC constructs, the reviewer feels that more extensive analyses under a physiological context are needed to support the conclusion.

Answer: We have now included a background of BRET assay in the results, and we validate the conclusion obtained with BRET data with microscopy as shown in Figure 3. The new text of the results reads as follow (page 6):

“... we used the BRET technique to monitor the distance between the donor Luc and the receptor YFP within the speck at real-time. The BRET signal is higher when these two epitopes are closer to each other and is lower when they are further apart (Compan et al., 2012a).”

3. Why was staining of ASC specks by the Luc antibodies more intense after the treatment, while ASC-YFP fluorescence remained constant whatever the treatment? A better control for this experiment is to compare the YFP fluorescence intensity with that of staining with an anti-YFP antibody.

Answer: The YFP emission is independent of the ASC conformation, as excitation will excite all YFP molecules independently of ASC conformation. Meanwhile antibody staining will result of the accessibility of the epitope that could be different after the treatment due to conformational changes of the structure. As suggested by the reviewer and as a control we show staining with anti-ASC antibody which is shown in Figure 4B. We do not preformed staining with anti-YFP to not interfere with excitation/emission of YFP protein.

In Fig. 3E, why RMFI of human ASC increased upon hypotonic stimulation in 3 min but decreased markedly after 20 min, which is inconsistent with the data in Fig. 3B? Also, there seems to be quite a big difference between mouse and human ASC when comparing the Fig. 3D and 3E data. What is the explanation for this discrepancy?

Answer: We agree with the reviewer that the response between mouse and human ASC is different after cell swelling, however while human ASC RMFI decrease (Figure 3E), the signal still significantly higher than control cells. The anti-ASC Ab used is anti-human ASC, so it is possible that the different affinity of the Ab for mouse and human ASC could explain the difference, together with the fact that the structural change of human ASC is bigger at short times when compared to mouse and at 20 min the RMFI of mouse and human are closer. In any case, we think this is a minor issue as the important point in this figure is that there is a change in ASC staining by Abs after stimulation. We have added a sentence in results to clarify this point (page 7):

“However, and differentially to mouse ASC, human ASC staining decreased after 20 min of hypotonic stimulation, but still significantly higher than in control conditions (Fig. 3B,E). This could be due to differences between mouse and human ASC speck structure or due to the specificity of the antibody against the human-ASC CARD domain”.

4. In Fig. 6, TcdB-induced ASC oligomerization was not that obvious. In fact, TcdB is a highly potent trigger of the pyrin inflammasome and can robustly induce ASC foci formation. Is there a technical issue here? Besides, the authors should investigate whether the structural change of ASC specks also participates in other inflammasome pathways.

Answer: To study TcdB-induced ASC speck (1h TcdB stimulation), we do not prime the macrophages with LPS to avoid NLRP3 inflammasome activation (as initial K^+ -efflux via gasdermin D pores could activate NLRP3 and sum up to Pyrin inflammasome). Therefore, the number of specking macrophages is smaller than in other studies that prime macrophages with LPS prior TcdB addition. As suggested by the reviewer, we

have now stimulated the NLRC4 inflammasome and found a similar structural change of the resulting ASC speck when nigericin was added (see new Figure 6D).

5. The authors showed that nigericin and hypotonic stimulation were not able to alter the number of endogenous ASC specks in macrophages upon TcdB stimulation. Does this suggest that the structural change of ASC specks after nigericin or hypotonic stimulation only occur in the artificial cell system and is dispensable for physiological caspase-1 activation?

Answer: We do not expect an increase in the number of ASC specks after nigericin stimulation (as the NLRP3 is not activated since no cell priming was performed). We report a change in the structure of the ASC oligomer, but this do not mean an increase of the number of ASC specks. In fact, in the recombinant cell system, we do not see an increase of ASC specks.

6. The authors propose that the initial low-level gasdermin D pore formation may positively amply the extent of inflammasome activation via the decrease of intracellular potassium concentration. This is an interesting idea of potential physiological significance. Can the authors provide experimental evidences supporting this?

Answer: Unfortunately, all our experimental attempts to evidence this failed (see new supplementary Fig S4). We found that the release of IL-1 β after Pysin inflammasome activation with CtdB in *Nlrp3*^{-/-} BMDMs was not modulated with either high K⁺ buffer or by nigericin. This could be due to a fast release of IL-1 β after activation of the inflammasome, resulting the modulation of the ASC specks by K⁺ efflux unable to further increase IL-1 β release. Therefore, we think that the K⁺ efflux-regulation of the ASC speck should be relevant at early time points of stimulation before IL-1 β release could be detected. Now we explain these results in the revised MS (page 10):

*“Despite the increased accessibility of the ASC^{CARD} domain within the ASC speck when intracellular K⁺ was decreased, we found that neither nigericin or the use of a buffer with a high K⁺ concentration were able to modulate IL-1 β release from *Nlrp3*^{-/-} macrophages (Fig. S4). This could be because when IL-1 β is detected in the supernatant there is already enough pyroptosis to saturate the system, and the modulation of the ASC speck by K⁺ efflux and further increase of caspase-1 activation, could not significantly change the final amount of released cytokine”.*

7. The language of the presentation needs to be significantly proved as there are many grammatic errors and awkward sentences.

Answer: We sent the text for English grammar and style correction.

Reviewer #2 (Comments to the Authors (Required)):

Comments on the study « ASC oligomer favor caspase-1CARD domain recruitment after intracellular potassium efflux » from Martin-Sanchez et al.,

In their study the authors address the importance of intracellular potassium (K⁺) concentration on ASC CARD domain accessibility to Caspase-1. These findings are worth investigating further, as so far, potassium has only been described as a central signal for the Nlrp3 inflammasome activation and assembly, but not for ASC-Caspase-1 interaction. The study is conducted with very interesting tools to detect intraASC oligomer structure. Yet, some parts seem to be a bit overstated in absence of proper experiments.

Point 1/

One puzzling thing to me is that the authors suggest that intracellular potassium is required for better Casp-1 recruitment to the oligomerized ASC in response to Pyn+potassium efflux triggering condition. Could the authors test whether inhibiting potassium efflux in their conditions alter the ASC speck intensity? But also, is potassium efflux altering the ASC-Casp-1 interaction after a single PYRIN inflammasome stimulation (meaning without any further stimulation with nigericin or hypotonic media)?

Answer: In Figure 6B we show that the ASC speck intensity relative to TcdB do not change when K⁺ efflux is prevented. Furthermore, we performed new experimental attempts to evidence a functional requirement of K⁺ efflux regulating Pyn inflammasome activity. However, we found that the release of IL-1 β after Pyn inflammasome activation with CtdB in *Nlrp3*^{-/-} BMDMs was not modulated with either high K⁺ buffer or by nigericin (see new supplementary Fig S4). This could be due to a fast release of IL-1 β after activation of the inflammasome, resulting the modulation of the ASC specks by K⁺ efflux unable to further increase IL-1 β release. Therefore, we think that the K⁺ efflux-regulation of the ASC speck should be relevant at early time points of stimulation before IL-1 β release could be detected. Now we explain these results in the revised MS (page 10):

*“Despite the increased accessibility of the ASC^{CARD} domain within the ASC speck when intracellular K⁺ was decreased, we found that neither nigericin or the use of a buffer with a high K⁺ concentration were able to modulate IL-1 β release from *Nlrp3*^{-/-} macrophages (Fig. S4). This could be because when IL-1 β is detected in the supernatant there is already enough pyroptosis to saturate the system, and the modulation of the ASC speck by K⁺ efflux and further increase of caspase-1 activation, could not significantly change the final amount of released cytokine”.*

Point 2/

But also, are the author findings also applying to the three other inflammasomes, namely NLRC4, CARD8 and NLRP1, where the ASC oligomerization status is different given they all express a CARD domain? One could speculate that intracellular [K⁺] might also regulate CARD-containing sensors and their direct, ASC-independent, Caspase-1 recruitment ? Could the authors test this ?

Answer: We have now stimulated the NLRC4 inflammasome in the presence of MCC950 (to block possible NLRP3 activation) and found a similar structural change of ASC speck when nigericin was added (see new Figure 6D).

Point 3/

A strong missing point is the lack of translation of the authors findings in inflammasome function. Meaning, could the author use K+ High [] or K+ low [] to translate their findings in Caspase-1 activity ? Gasdermin D cleavage ? Il1beta maturation and release ? Cell death ?

Answer: Unfortunately, and explained in the first point, all our experimental attempts to evidence this failed (see new supplementary Fig S4).

Reviewer #3 (Comments to the Authors (Required)):

Martin-Sanchez et al present an intriguing study on the role of intracellular potassium ion concentration in regulating the structure of the key inflammasome adapter protein ASC. Previous studies have shown that K+ efflux is essential for NLRP3 activation in response to almost all NLRP3 stimuli (except R837 and alternative NLRP3 activation in monocytes). The authors largely use a HEK cell overexpression system as their model (as they have in numerous previous studies). Using this system, they show that ASC can form functional specks even in the absence of an inflammasome sensor. Using a BRET biosensor approach, they then show that ASC undergoes a structural change when HEK cells are stimulated with nigericin or hypotonic medium (which are known to cause K+ efflux). They support their BRET observations with an antibody-based approach and show that in similar conditions (HEK cells stimulated with nigericin or hypotonic medium) there is a change in the binding ability of an antibody directed to the CARD domain of ASC - indicating a structural change in ASC. The authors then suggest that this structural change in ASC enhances caspase-1 binding to ASC (CARD-CARD interaction). In macrophages where the pyrin inflammasome has been activated, they can observe the change in ASC-CARD antibody binding when cells are also treated with nigericin or hypotonic medium. Overall the authors conclude that potassium efflux not only regulates NLRP3 but also the structure of the ASC speck and thus will influence signalling from other inflammasomes such as pyrin and AIM2.

While the data presented here are sound, and I believe that this is potentially an important and novel observation about the cell biology of inflammasomes, I think that more experiments are needed to support the conclusions drawn by the authors. My major and minor comments are listed below but for clarity the two keys questions that need to be resolved/addressed are:

1. Whether there is any role for other ions such as chloride in this phenotype.

Answer: In this study we centered on the role of K^+ efflux, as this ion flux is important for NLRP3 inflammasome activation. Therefore, it is reasonable that other ions (as Ca^{2+} , Na^+ or Cl^- as the reviewer point) could be important in modulating the structure of the ASC speck. In fact, studies in collaboration with Borugh's lab demonstrate that Cl^- efflux was important for ASC speck formation downstream NLRP3 inflammasome activation (PMID: 30232264). Although we agree with the importance of the reviewer comment, we feel that addressing the role of other ions in the structure of ASC oligomers would be a matter of a future study.

2. Whether the increased availability of the CARD domain has any influence on caspase-1 activation particularly in an immune cell such as a macrophage.

Unfortunately, all our experimental attempts to evidence this failed (see new supplementary Fig S4). We found that the release of IL-1 β after Pyrin inflammasome activation with CtdB in *Nlrp3*^{-/-} BMDMs was not modulated with either high K^+ buffer or by nigericin. This could be due to a fast release of IL-1 β after activation of the inflammasome, resulting the modulation of the ASC specks by K^+ efflux unable to further increase IL-1 β release. Therefore, we think that the K^+ efflux-regulation of the ASC speck should be relevant at early time points of stimulation before IL-1 β release could be detected. Now we explain these results in the revised MS (page 10):

*“Despite the increased accessibility of the ASC^{CARD} domain within the ASC speck when intracellular K^+ was decreased, we found that neither nigericin or the use of a buffer with a high K^+ concentration were able to modulate IL-1 β release from *Nlrp3*^{-/-} macrophages (Fig. S4). This could be because when IL-1 β is detected in the supernatant there is already enough pyroptosis to saturate the system, and the modulation of the ASC speck by K^+ efflux and further increase of caspase-1 activation, could not significantly change the final amount of released cytokine”.*

Major Comments

General

1. A diagram that shows how the authors think the ASC structure changes would be very helpful.

Answer: As requested, we added a possible diagram in Supplementary Figure S4B.

Introduction

2. The introduction is missing some key references (it is not sufficient to just cite reviews 1 and 2) which must be included and discussed to provide the correct scientific context for this study (the concept of ion flux regulating caspase-1 activation has been around for decades). Changes in intracellular potassium were first linked to IL-1beta processing by Perregaux and Gabel (PMID: 8195155. J Biol Chem. 1994 May 27;269(21):15195-203. Interleukin-1 beta maturation and release in response to ATP and nigericin.

Evidence that potassium depletion mediated by these agents is a necessary and common feature of their activity.) The activation of the NLRP3 inflammasome by low K⁺ was first reported by Petrilli et al (Pétrilli V, Papin S, Dostert C, Mayor A, Martinon F, Tschopp J. Cell Death Differ. 2007 Sep;14(9):1583-9. Epub 2007 Jun 29. Activation of the NALP3 inflammasome is triggered by low intracellular potassium concentration. PMID: 17599094)

Answer: We thank the reviewer for this annotation, we have included these references in the introduction.

Results

3. Figure 1A - K⁺ has not been measured here. This could be attributed to cell swelling or changes in other ions such as Cl⁻. The changes in intracellular K⁺ levels (that are assumed to occur) should actually be demonstrated in this HEK system.

Answer: The reviewer is right, now we have measure that there is a decrease of intracellular K⁺ after cell swelling or nigericin treatment (see new Fig S1A).

4. Fig S1A-B (and Fig. 2E, S2C) - Although a high K⁺ buffer blocks these effects the specific details of this buffer haven't been provided in the methods. I would presume this contains a high concentration of KCl so could the effect be due to high Cl⁻? High extracellular Cl⁻ has been shown to inhibit NLRP3 activation (Tang et al 2017 PMID: 28779175 and Domingo-Fernandez et al 2014 PMID: 28576828).

Answer: We are sorry for not describing high K⁺ buffer composition, now it is described in the material and methods section (page 16), and the buffer has 147 mM of KCl, but this was compensated with a proportional reduction of NaCl concentration to maintain both osmolarity and Cl⁻ concentration. Therefore, the actual concentration of Cl⁻ is the same that the normal buffer and do not have a high Cl⁻ concentration.

5. Fig 1E This assay shows that caspase-1 autocatalytic cleavage is occurring. It would be more convincing if an additional assay was included to demonstrate caspase-1 activity such as staining with FAM-YVAD-FMK or by measuring substrate processing (e.g. co-transfection of a pro-IL-1beta expressing plasmid and measure IL-1beta cleavage).

Answer: We have now show that pro-IL-1β is being cleavage in the recombinant cellular system, suggesting active caspase-1 (see new panel in Figure 1E).

6. Fig 1G and Fig 1H are the filaments we see in these two images supposed to be similar or different? The high-magnification image of the ASC structure in Fig 1G is very compelling. Could this be used to analyse the same structures shown in Fig 1H with ASC-RFP?

Answer: The filaments shown in figure 1H are the result of using a Nb against the CARD domain of ASC, that can block ASC speck formation (PMID: 27069117). Therefore, the speck of Fig 1G is the result of ASC filament interaction via CARD-CARD domain interactions and probably a different structure. The detailed analysis of the ASC filament structure is out of the scope of the present study and has been recently addressed (doi.org/10.1101/2021.09.17.460822). Now, we show that ASC-Luc tagged is also able to form ASC speck with a filamentous nature (see new panel in Fig 1G).

7. Fig S2A, Fig 2A, Fig 3A: ASC-luciferase specks appear to be bigger and on the outside of the co-expressed specks while ASC-YFP is denser more concentrated. Why might this be?

Answer: The ASC speck is a compact structure, mainly in the core, so the anti-Luc Ab used to stain ASC-Luc would not be able to access the epitopes in the compact core and stain properly the inner, therefore the anti-Luc fluorescence for ASC-Luc is mainly in the outside of the ASC speck. However, ASC-YFP emit fluorescence in all the ASC speck, but because the core is denser (and therefore with more ASC molecules), the signal appears concentrated in the core and weaker in the edge. Figure 1G gives an idea of this structure with a denser and more compact core and with a branched and less compact edge of the speck.

8. Figure 3A,B - ASC-YFP intensity should also be quantified here as it looks like there is increased fluorescence after 20 min nigericin. Is it possible that the nigericin and hypotonic treatments non-specifically affect YFP and/or luciferase? It might be useful to include a control where a different protein (i.e. not ASC) is tagged with luciferase and/or RFP to exclude this possibility.

Answer: We have now quantified ASC-YFP intensity for the analyzed specks, and we found that both nigericin or hypotonic treatments do not affect YFP signal (see new Figure S3A). The expression of another protein with YFP or RFP will not result in a compact speck-like structure and therefore would be not possible to do quantification of a speck like structure.

9. Figure 3 'These data suggest that upon K⁺ efflux, the ASC CARD domain is more accessible to antibody staining '. I think this is an overinterpretation of these results. For example, the antibody staining could be enhanced because of the plasma membrane changes caused by nigericin treatment.

Answer: We thanks the reviewer for this comment, we also think other explanations are possible, that is why we use the term “suggest” and not “prove”. The explanation provided by the reviewer (nigericin induced plasma membrane permeabilization) cannot be suggested because: (1) nigericin do not induce plasma membrane permeabilization in HEK293 cells, as they do not have caspase-1/gasdermin D to permeabilize the membrane; and (2) before Ab staining cells are permeabilized with detergent, so similar accessibility of Ab is expected after the different treatments.

10. Fig. 4A, S3C - I'm not sure it's correct that this antibody can only detect oligomerised ASC. For example, in PMID: 27221487 Fig 3B there is clear staining non-oligomerised ASC using the Biologend antibody (HASC-71) - is this difference potentially due to antibody concentration?

Answer: In the mentioned chapter (PMID: 27221487) the authors use human PBMCs. As we indicate in the Table S1, the Biologend Ab can recognize human ASC in both the soluble and oligomeric forms, but the mouse ASC is only detected when form oligomeric specks. In Fig. 4A and S3C we use mouse ASC-YFP, and the Biologend in this case only stain after stimulation and speck formation. A new text has been added to the results section to clarify this (page 8):

“Of note, and contrary to the mouse ASC, the Biologend antibody against the ASCCARD can stain both soluble and oligomeric human ASC (Table S1)”.

11. Figure 6 - TcdB stimulation can also activate other signalling pathways (e.g. MAPK - see review PMID: 27108093) so there is the potential that it could prime NLRP3. Another inflammasome should be tested - I would suggest AIM2 activation which does not require priming.

Answer: We have now stimulated the NLRC4 inflammasome in the presence of MCC950 (to block possible NLRP3 activation) and found a similar structural change of ASC speck when nigericin was added (see new Figure 6D).

12. Figure 6B-C the percentage change in ASC-CARD staining is very small here and doesn't seem to be consistent with the representative images shown in 6C (i.e 1-1.4% vs 707-3202). This should be presented alongside data from cells where NLRP3 has been activated for comparison.

Answer: If we stimulate NLRP3 with nigericin to measure the intensity of stained ASC specks, we would not have a control group also with nigericin to compare with. In figure 6B we made the average of 179 ASC specks, and in the picture of figure 6C we represented only one of them to show the staining differences. Now, we also show how nigericin could modulate NLRC4 induced ASC specks (see new Figure 6D).

13. Figure 6B-C The authors need to test if this observation results in enhanced caspase-1 activation. For example, by quantifying caspase-1 activity using a fluorescent substrate such as FAM-YVAD-FMK or by measuring caspase-1 substrate cleavage such as IL-1beta, IL-18 or gasdermin-D by ELISA or Western blotting.

Answer: Unfortunately, all our experimental attempts to evidence this failed (see new supplementary Fig S4). We found that the release of IL-1 β after Pyrin inflammasome activation with CtdB in *Nlrp3*^{-/-} BMDMs was not modulated with either high K⁺ buffer or by nigericin. This could be due to a fast release of IL-1 β after activation of the

inflammasome, resulting the modulation of the ASC specks by K^+ efflux unable to further increase IL-1 β release. Therefore, we think that the K^+ efflux-regulation of the ASC speck should be relevant at early time points of stimulation before IL-1 β release could be detected. Now we explain these results in the revised MS (page 10):

“Despite the increased accessibility of the ASC^{CARD} domain within the ASC speck when intracellular K^+ was decreased, we found that neither nigericin or the use of a buffer with a high K^+ concentration were able to modulate IL-1 β release from Nlrp3^{-/-} macrophages (Fig. S4). This could be because when IL-1 β is detected in the supernatant there is already enough pyroptosis to saturate the system, and the modulation of the ASC speck by K^+ efflux and further increase of caspase-1 activation, could not significantly change the final amount of released cytokine”.

Minor Comments

General

1. Some editing of the grammar would enhance the clarity of the manuscript.

Answer: We sent the text for English grammar and style correction.

Introduction

2. Line 2/3: 'NLRP3 is a multiprotein complex, whose oligomerization occurs' suggest change to 'The NLRP3 inflammasome is a multiprotein complex whose activation occurs...' or 'NLRP3 oligomerises...' Just to be clear that NLRP3 itself is not a multiprotein complex.

Answer: We thanks the reviewer for this observation, we made the suggested change.

Results

3. Figure S1C - Based on these images there does look like there is less D303N oligomerisation in the 90 mOsm treatment. Perhaps this could be quantified or other representative images shown.

Answer: We agree with the reviewer and after checking different pictures, seems that there is less puncta for NLRP3 D303N when cells were incubated in hypotonic solution. Although this is an interesting observation, we do not think quantification would help the whole story, since our study do not center on the CAPS NLRP3 puncta formation. Even if we quantify this decrease, we still have no idea on the mechanism behind this observation. Therefore, to keep the story focused on ASC specks, we decided not to include a quantification of this phenomenon.

4. Western blots should have molecular weight markers added.

Answer: We have now included molecular weight markers for the Western blots.

5. Some of the scale bars on the microscopy images are not that visible - could this be made clearer with a thicker or larger line?

Answer: We have made scale bars thicker.

6. Fig S2A is missing scale bars.

Answer: We have added scale bars for this figure panel.

7. Fig 2G - a legend showing what the block dots and grey dots are would be really helpful for the reader here.

Answer: As suggested by the reviewer we have added a legend to this panel.

8. Figures 3 C-E - I think it would be useful to show some representative images here. They could be included as a supplemental figure.

Answer: We have now included representative images in Figure 3F.

9. Figure S3B - the representative images are not very clear. If possible, a higher magnification i.e. the same as the magnification in S3A should be presented.

Answer: We have increased the magnification of the pictures of Figure S3B.

10. Figure 4B and S3D-E are these data from the same experiments? It would be clearer to show the data from the high K⁺ on the same graph.

Answer: Yes, this data is from the same experiment, as suggested now we show data from high K⁺ in the same graph (Figure 4B).

11. Figure 5A Why is there so much less CARD-EGFP staining? Why was ten-fold less protein used?

Answer: In our initial experiments we tested different dilutions for ASC-RFP and CARD-EGFP. We found that ASC-RFP was much brighter than CARD-EGFP protein, and the staining was saturated much quicker with ASC-RFP than when CARD-EGFP was used. This is why we used 10-fold less ASC-RFP than CARD-EGFP, since with that differential dilutions we were not saturating the preps.

12. Figure 5C What is the control condition here? Is it untreated cells (as shown in 5A)?

Answer: The control condition is 20 min with no stimulation (isotonic solution). We now indicated this in the figure legend.

April 3, 2023

RE: JCB Manuscript #202003053R

Dr. Pablo Pelegrin
Instituto Murciano de Investigación Biosanitaria
Carretera Buenavista
Murcia 30120
Spain

Dear Dr. Pelegrin:

Thank you for submitting your revised manuscript entitled "ASC oligomer favor caspase-1 CARD domain recruitment after intracellular potassium efflux". The original reviewers have now assessed your revised manuscript and, as you can see, they are overall satisfied with the revisions. However, revs #2 and #3 note that the relevance of ASC specks formation in inflammasome signaling still remains unclear. Thus, we would like to suggest that you please revise the abstract and text accordingly. We would be happy to publish your paper in JCB pending final revisions necessary to address these minor reviewers' comments. Please, when submitting the final revision, make sure that you comply with our formatting guidelines (see details below).

To avoid unnecessary delays in the acceptance and publication of your paper, please read the following information carefully. Please go through all the formatting points paying special attention to those marked with asterisks.

A. MANUSCRIPT ORGANIZATION AND FORMATTING:

Full guidelines are available on our Instructions for Authors page, <https://jcb.rupress.org/submission-guidelines#revised>.
Submission of a paper that does not conform to JCB guidelines will delay the acceptance of your manuscript.

1) Text limits: Character count for Articles and Tools is < 40,000, not including spaces. Count includes title page, abstract, introduction, results, discussion, and acknowledgments. Count does not include materials and methods, figure legends, references, tables, or supplemental legends.

2) Figures limits: Articles and Tools may have up to 10 main text figures.

*** Please note that main text figures should be provided as individual, editable files.

3) Figure formatting:

Molecular weight or nucleic acid size markers must be included on all gel electrophoresis.

Scale bars must be present on all microscopy images, including inset magnifications.

Also, please avoid pairing red and green for images and graphs to ensure legibility for color-blind readers. If red and green are paired for images, please ensure that the particular red and green hues used in micrographs are distinctive with any of the colorblind types. If not, please modify colors accordingly or provide separate images of the individual channels.

4) Statistical analysis:

*** Error bars on graphic representations of numerical data must be clearly described in the figure legend.

*** The number of independent data points (n) represented in a graph must be indicated in the legend. Please, indicate whether N refers to technical or biological replicates (i.e. number of analyzed cells, samples or animals, number of independent experiments).

If independent experiments with multiple biological replicates have been performed, we recommend using distribution-reproducibility SuperPlots (please, see Lord et al., JCB 2020) to better display the distribution of the entire dataset, and report statistics (such as means, error bars, and P values) that address the reproducibility of the findings.

Statistical methods should be explained in full in the materials and methods in a separate section.

For figures presenting pooled data the statistical measure should be defined in the figure legends.

*** Please also be sure to indicate the statistical tests used in each of your experiments (both in the figure legend itself and in a separate methods section) as well as the parameters of the test (for example, if you ran a t-test, please indicate if it was one- or two-sided, etc.).

If you used parametric tests in your study (i.e. t-tests), you should have first determined whether the data was normally distributed before selecting that test. In the stats section of the methods, please indicate how you tested for normality. If you did not test for normality, you must state something to the effect that "Data distribution was assumed to be normal but this was not formally tested."

5) Abstract and title:

The abstract should be no longer than 160 words and should communicate the significance of the paper for a general audience.

The title should be less than 100 characters including spaces. Make the title concise but accessible to a general readership.

6) Materials and methods:

*** Should be comprehensive and not simply reference a previous publication for details on how an experiment was performed. The text should not refer to methods "...as previously described."

Also, the materials and methods should be included in the main manuscript text and not in the supplementary materials.

7) For all cell lines, vectors, constructs/cDNAs, etc. -all genetic material: please include database / vendor ID (e.g., Addgene, ATCC, etc.) or if unavailable, please briefly describe their basic genetic features, even if described in other published work or gifted to you by other investigators (and provide references where appropriate).

Please be sure to provide the sequences for all of your oligos: primers, si/shRNA, RNAi, gRNAs, etc. in the materials and methods.

*** You must also indicate in the methods the source, species, and catalog numbers/vendor identifiers (where appropriate) for all your antibodies, including secondary, and the system used to collect the signal from the antibodies. If antibodies are not commercial, please add a reference citation if possible. Please add the catalog numbers for all your antibodies and the system/machine used to acquire the signal.

8) Microscope image acquisition:

The following information must be provided about the acquisition and processing of images:

- a. Make and model of microscope
- b. Type, magnification, and numerical aperture of the objective lenses
- c. Temperature
- d. imaging medium
- e. Fluorochromes
- f. Camera make and model
- g. Acquisition software
- h. Any software used for image processing subsequent to data acquisition. Please include details and types of operations involved (e.g., type of deconvolution, 3D reconstitutions, surface or volume rendering, gamma adjustments, etc.).

10) Supplemental materials:

There are strict limits on the allowable amount of supplemental data. Articles and Tools may have up to 5 supplemental figures. There is no limit for supplemental tables.

*** Please note that supplemental figures and tables should be provided as individual, editable files.

*** A summary of all supplemental material should appear at the end of the Materials and Methods section (please see any recent JCB paper for an example of this summary).

11) Video legends:

Video legends should describe what is being shown, the cell type or tissue being viewed (including relevant cell treatments, concentration and duration, or transfection), the imaging method (e.g., time-lapse epifluorescence microscopy), what each color represents, how often frames were collected, the frames/second display rate, and the number of any figure that has related video stills or images.

12) eTOC summary:

*** A ~40-50 word summary that describes the context and significance of the findings for a general readership should be included on the title page. The statement should be written in the present tense and refer to the work in the third person. It should begin with "First author name(s) et al..." to match our preferred style.

13) Conflict of interest statement:

JCB requires inclusion of a statement in the acknowledgements regarding competing financial interests. If no competing financial interests exist, please include the following statement: "The authors declare no competing financial interests."

14) Author contribution:

A separate author contribution section is required following the Acknowledgments in all research manuscripts.

*** All authors should be mentioned and designated by their first and middle initials and full surnames and the CRediT nomenclature is encouraged (<https://casrai.org/credit/>).

15) ORCID IDs: ORCID IDs are unique identifiers allowing researchers to create a record of their various scholarly contributions in a single place. At resubmission of your final files, please consider providing an ORCID ID for as many contributing authors as possible.

16) Materials and data sharing:

All animal and human studies must be conducted in compliance with relevant local guidelines, such as the US Department of Health and Human Services Guide for the Care and Use of Laboratory Animals or MRC guidelines, and must be approved by the authors' Institutional Review Board(s). A statement to this effect with the name of the approving IRB(s) must be included in the Materials and Methods section.

*** Journal of Cell Biology now requires a data availability statement for all research article submissions. These statements will be published in the article directly above the Acknowledgments. The statement should address all data underlying the research presented in the manuscript. Please visit the JCB instructions for authors for guidelines and examples of statements at (<https://rupress.org/jcb/pages/editorial-policies#data-availability-statement>).

All datasets included in the manuscript must be available from the date of online publication, and the source code for all custom computational methods, apart from commercial software programs, must be made available either in a publicly available database or as supplemental materials hosted on the journal website. Numerous resources exist for data storage and sharing (see Data Deposition: <https://rupress.org/jcb/pages/data-deposition>), and you should choose the most appropriate venue based on your data type and/or community standard. If no appropriate specific database exists, please deposit your data to an appropriate publicly available database.

17) Please note that JCB now requires authors to submit Source Data used to generate figures containing gels and Western blots with all revised manuscripts. This Source Data consists of fully uncropped and unprocessed images for each gel/blot displayed in the main and supplemental figures. The Source Data files will be directly linked to specific figures in the published article.

As your paper includes cropped gel and/or blot images, please be sure to provide one Source Data file for each figure that contains gels and/or blots along with your revised manuscript files. File names for Source Data figures should be alphanumeric without any spaces or special characters (i.e., SourceDataF#, where F# refers to the associated main figure number or SourceDataFS# for those associated with Supplementary figures). The lanes of the gels/blots should be labeled as they are in the associated figure, the place where cropping was applied should be marked (with a box), and molecular weight/size standards should be labeled wherever possible.

Source Data Figures should be provided as individual PDF files (one file per figure). Authors should endeavor to retain a

minimum resolution of 300 dpi or pixels per inch. Please review our instructions for export from Photoshop, Illustrator, and PowerPoint here: <https://rupress.org/jcb/pages/submission-guidelines#revised>

B. FINAL FILES:

Thank you for this interesting contribution, we look forward to publishing your paper in Journal of Cell Biology.

Sincerely,

Craig Roy
Monitoring Editor
Journal of Cell Biology

Lucia Morgado-Palacin, PhD
Scientific Editor
Journal of Cell Biology

Reviewer #1 (Comments to the Authors (Required)):

I am happy with the revision. The revised manuscript is a nice addition to the mechanistic understanding of ASC-containing (NLRP3) inflammasome activation.

Reviewer #2 (Comments to the Authors (Required)):

In this work, the Pelegrin group describes an intriguing process by which potassium efflux supports ASC-dependent Caspase-1 recruitment for subsequent activation. This is of importance as, potassium efflux remains an enigmatic regulator of the NLRP3

inflammasome, but its role at directly regulating a general inflammasome response has never been addressed.

The revised version of the manuscript as well as the response to the reviewer's comments/suggestions has been strongly addressed either by discussing various raised issues or experimentally. Furthermore, in its revised form the study is of nice interest for the field.

All in one, we are in favor of article acceptance.

Reviewer #3 (Comments to the Authors (Required)):

In their revised manuscript the authors have clarified many small issues that were raised and have added a few experiments that have addressed a lot of my comments. These include measuring intracellular potassium (Fig S1A), IL-1beta blots (Fig 1B), YFP fluorescence measurements (Fig 3A, S3A), IL-1beta measurements following Pypin stimulation with nigericin (Fig S4A), and ASC speck intensity after NLRC4 activation with nigericin (Fig 6D). I really appreciate that adding additional experiments to this study was very challenging given the global pandemic. The manuscript very nicely shows that changes in potassium alter the conformation of ASC specks which I think will be very interesting for the field. However, whether these alterations have any impact on inflammasome signalling i.e., caspase-1 activation and IL-1beta processing or pyroptosis is still unclear. Indeed, the authors have tried to address this but have not been able to see any differences in IL-1beta secretion. I feel that this lack of effect on inflammasome signalling somewhat undermines the importance of the observations and I don't think that they have demonstrated that low intracellular potassium "enhances the recruitment of procaspase-1 through ASC specks formed from different inflammasomes" as stated in the abstract.

Pablo Pelegrín, PhD
Scientific Deputy Director & Principal Investigator
Biomedical Research Institute of Murcia
University of Murcia
30120 Murcia (Spain)

April 21, 2023

Dear Dr Morgado-Palacin, Scientific Editor of Journal of Cell Biology,

Thank you very much to allow us to resubmit a revised version of our manuscript entitled 'ASC oligomer favor caspase-1 CARD domain recruitment after intracellular potassium efflux' (#202003053R) for consideration in JBC.

In this revised version, we have implemented journal formatting and addressed reviewer #2 and #3 concern about the relevance of ASC specks formation in inflammasome signaling. We have therefore changed the abstract and the main text to make clear that the changes reported in the ASC oligomeric structure seems not to directly impact on inflammasome signaling.

Below you will find a detailed response addressing all the reviewers' comments point by point. We hope that our modifications would satisfy the point raised by the reviewer #3,

Thank you very much for your consideration.
Yours faithfully,

Pablo Pelegrín on behalf of all authors.

Point by point rebuttal letter:

JCB manuscript #202003053R

Manuscript entitled "ASC oligomer favor caspase-1 CARD domain recruitment after intracellular potassium efflux".

Reviewer #1 (Comments to the Authors (Required)):

I am happy with the revision. The revised manuscript is a nice addition to the mechanistic understanding of ASC-containing (NLRP3) inflammasome activation.

Answer: We thanks the reviewer for his/her comment.

Reviewer #2 (Comments to the Authors (Required)):

In this work, the Pelegrin group describes an intriguing proces by which potassium efflux supports ASC-depdent CApase-1 recruitment for subsequent activation. This is of importance as, potassium efflux remains an enigmatic regulator of the NLRP3 inflammasome, but its role at directly regulating a general inflammasome response has never been adressed.

The revised version of the manuscript as well as the response to the reviewer's comments/suggestions has been strongly adressed either by discussing various raised issues or experimentally. Firthermore, in its revised form the study is of nice interest for the field.

All in one, we are in favor of article acceptance.

Answer: We thanks the reviewer for his/her comment, we soften the text regarding the conclusion on the direct regulation of inflammasome by K⁺ efflux.

Reviewer #3 (Comments to the Authors (Required)):

In their revised manuscript the authors have clarified many small issues that were raised and have added a few experiments that have addressed a lot of my comments. These include measuring intracellular potassium (Fig S1A), IL-1beta blots (Fig 1B), YFP fluorescence measurements (Fig 3A, S3A), IL-1beta measurements following Pyn stimulation with nigericin (Fig S4A), and ASC speck intensity after NLRC4 activation with nigericin (Fig 6D). I really appreciate that adding additional experiments to this study was very challenging given the global pandemic. The manuscript very nicely shows that changes in potassium alter the conformation of ASC specks which I think will be very interesting for the field. However, whether these alterations have any impact on inflammasome signalling i.e., caspase-1 activation and IL-1beta processing or pyroptosis is still unclear. Indeed, the authors have tried to address this but have not been able to see any differences in IL-1beta secretion. I feel that this lack of effect on inflammasome signalling somewhat undermines the importance of the observations and I don't think that they have demonstrated that low intracellular potassium "enhances the recruitment

of procaspase-1 through ASC specks formed from different inflammasomes" as stated in the abstract.

Answer: We agree with the reviewer comment, and we have changed the text (including the abstract) to clarify that our data demonstrate that the structural change found in the ASC oligomers do not have a potential impact on inflammasome signaling.